# Coordinated control of adiposity and growth by anti-anabolic kinase ERK7

Kiran Hasygar[1,2], Onur Deniz[1,2], Ying Liu[1,2], Josef Gullmets[1,2], Riikka Hynynen[1,2], Hanna Ruhanen[1,3], Krista Kokki[1,2], Reijo Käkelä[1,3] & Ville Hietakangas[1,2,*] [iD]

## Abstract

Energy storage and growth are coordinated in response to nutrient status of animals. How nutrient-regulated signaling pathways control these processes *in vivo* remains insufficiently understood. Here, we establish an atypical MAP kinase, ERK7, as an inhibitor of adiposity and growth in *Drosophila*. *ERK7* mutant larvae display elevated triacylglycerol (TAG) stores and accelerated growth rate, while overexpressed ERK7 is sufficient to inhibit lipid storage and growth. ERK7 expression is elevated upon fasting and *ERK7* mutant larvae display impaired survival during nutrient deprivation. ERK7 acts in the fat body, the insect counterpart of liver and adipose tissue, where it controls the subcellular localization of chromatin-binding protein PWP1, a growth-promoting downstream effector of mTOR. PWP1 maintains the expression of *sugarbabe*, encoding a lipogenic Gli-similar family transcription factor. Both PWP1 and Sugarbabe are necessary for the increased growth and adiposity phenotypes of ERK7 loss-of-function animals. In conclusion, ERK7 is an anti-anabolic kinase that inhibits lipid storage and growth while promoting survival on fasting conditions.

**Keywords** growth; lipid metabolism; nutrient sensing; signaling
**Subject Categories** Development; Metabolism

## Introduction

Animals live in changing nutrient landscape and are competing on limited nutrient resources. These conditions set requirements for robust physiological control, allowing efficient growth and energy storage on abundant nutrition and rapid attenuation of anabolic processes upon nutrient deprivation. How metabolic pathway activities are controlled in peripheral tissues of animals to achieve coordinated control of growth and energy storage, remains insufficiently understood. The fat body (functional counterpart of mammalian liver

and adipose tissue) has emerged as a key coordinator of growth signaling in *Drosophila* (Boulan *et al*, 2015) along with its essential role as a lipid synthesis and storage organ (Heier & Kuhnlein, 2018). Thus, it is a signaling hub for nutrient-dependent homeostatic control.

Nutrient-responsive physiological control relies on nutrient-sensing signaling pathways and gene regulatory networks, which respond to specific groups of macronutrients and drive anabolic processes upon high nutrient intake. For example, amino acid-responsive mTOR complex 1 (mTORC1) promotes protein and lipid biosynthesis and is a major driver of organismal growth (Oldham *et al*, 2000; Zhang *et al*, 2000; Colombani *et al*, 2003; Porstmann *et al*, 2008; Saxton & Sabatini, 2017). A key target process for mTORC1-dependent growth control is ribosome biogenesis (Mayer *et al*, 2004; Grewal *et al*, 2007; Teleman *et al*, 2008; Marshall *et al*, 2012). We recently identified chromatin-binding protein PWP1 as an mTORC1-dependent regulator of growth through its function as a positive regulator of RNA polymerase I- and III-mediated ribosomal RNA (rRNA) expression (Liu *et al*, 2017; Liu *et al*, 2018). Inhibition of rRNA expression in the fat body, either by nutrient deprivation or genetically, leads to prominent growth inhibition (Grewal *et al*, 2007; Marshall *et al*, 2012; Liu *et al*, 2017). Intracellular sugars are sensed by a conserved transcription factor (TF) heterodimer Mondo-Mlx, which promotes *de novo* lipogenesis in the fat body by activation of lipid biosynthetic genes (Sassu *et al*, 2012; Havula *et al*, 2013; Musselman *et al*, 2013; Mattila *et al*, 2015). In addition to direct regulation of metabolic genes, Mondo-Mlx controls downstream TFs, such as Gli-similar 2 ortholog Sugarbabe, which acts as a feed-forward activator of lipid biosynthesis upon sustained sugar feeding (Mattila *et al*, 2015; Luis *et al*, 2016). The anabolic nutrient-sensing pathways are counterbalanced by anti-anabolic kinases, including the AMP-activated protein kinase (AMPK) (Herzig & Shaw, 2018). *Drosophila* AMPK is necessary for larval growth and adiposity by affecting visceral muscle function and food intake, rather than acting tissue autonomously in the fat body (Bland *et al*, 2010). This implies that other anti-anabolic kinases might be involved in fat body-mediated homeostatic control during the growth phase of the animal.

ERK7 [also known as ERK8, MAPK15 (Coulombe & Meloche, 2007)] is an atypical MAP kinase implicated in a number of cellular processes, including proliferation (Iavarone *et al*, 2006; Xu *et al*,

1 Molecular and Integrative Biosciences Research Programme, Faculty of Biological and Environmental Sciences, University of Helsinki, Helsinki, Finland
2 Institute of Biotechnology, University of Helsinki, Helsinki, Finland
3 Helsinki University Lipidomics Unit (HiLIPID), Helsinki Institute for Life Science (HiLIFE) and Biocenter Finland, Helsinki, Finland
 *Corresponding author. Tel: +35 8503 182927; E-mail: ville.hietakangas@helsinki.fi

2010; Chia *et al*, 2014; Colecchia *et al*, 2015; Jin *et al*, 2015; Rossi *et al*, 2016), secretion (Zacharogianni *et al*, 2011; Hasygar & Hietakangas, 2014), and genomic integrity (Groehler & Lannigan, 2010; Cerone *et al*, 2011; Rossi *et al*, 2016). Moreover, several independent lines of evidence suggest a putative role for ERK7 in nutrient-responsive homeostatic control. Studies in cultured *Drosophila* and mammalian cells have shown upregulation of ERK7 protein expression upon amino acid deprivation (Zacharogianni *et al*, 2011; Colecchia *et al*, 2012). Upregulation of ERK7 expression leads to kinase autoactivation, as ERK7 lacks the upstream activating kinases of canonical MAPK pathways (Abe *et al*, 2001; Klevernic *et al*, 2006). ERK7 activation in cultured mammalian cells promotes macroautophagy, an adaptive mechanism to amino acid shortage to restore cellular amino acid pools (Colecchia *et al*, 2012; Colecchia *et al*, 2018). Our earlier study showed fasting-induced upregulation of *erk7* gene expression in the insulin-producing cells (IPCs) of *Drosophila* larvae (Hasygar & Hietakangas, 2014). Activation of ERK7 in the IPCs suppresses secretion of insulin-like peptides leading to inhibition of larval growth, a response observed upon nutrient deprivation (Hasygar & Hietakangas, 2014). Moreover, genome-wide association studies (GWAS) have suggested that variants of the human ortholog of *ERK7* are associated with obesity (Li *et al*, 2012). Despite these pieces of evidence pointing to a role for ERK7 in nutrient signaling, its physiological role in metabolic control has not been previously addressed.

Here, we provide evidence that ERK7 acts as an anti-anabolic regulator *in vivo* by inhibiting lipid storage and growth in *Drosophila* larvae. ERK7 acts tissue autonomously in the fat body, where it counteracts the downstream effectors of major anabolic nutrient sensors, mTOR and Mondo-Mlx. Specifically, ERK7 regulates the subcellular localization of chromatin-binding protein PWP1 and, consequently, downregulates the expression of the lipogenic TF gene *sugarbabe*. Through these downstream effectors, ERK7 downregulates anabolic processes that promote lipid storage and growth.

## Results

### ERK7 inhibits lipid storage in the fat body

In order to explore the physiological role of ERK7, we used Crispr/Cas9 to generate a null mutant *Drosophila* allele, *ERK7*[1] (Figs 1A

and EV1A). Another line, which underwent the same screening process, but did not contain any mutations in the *ERK7* gene, was used as control. *ERK7*[1] animals were viable with no obvious morphological defects, but mutant males were fully infertile (Fig EV1B). Experiments with stage and density-controlled larvae showed that *ERK7*[1] mutants displayed significantly elevated triacylglycerol (TAG) levels (Fig 1B). To test if ERK7 regulates lipid storage tissue autonomously, we used fat body-specific (*CG-GAL4*) depletion of ERK7 by two independent RNAi lines, which phenocopied the *ERK7*[1] mutant (Figs 1C and EV2A). In line with the TAG measurements, *ERK7*[1] mutant fat body cells contained significantly higher numbers of lipid droplets when compared to control tissue, while the mean size of the droplets was not affected (Figs 1D and E, and EV2B). To assess the specific impact of ERK7 on lipid storage, we performed lipidomic analysis with larvae fed [13C]glucose and detected the ratio of newly synthesized [13C]TAG or unlabeled TAG to phosphatidylethanolamine (PE) (i.e., storage lipid/membrane lipid). These elevated newly synthesized and long-term TAG/PE ratios observed in the *ERK7*[1] mutant fat bodies (Fig 1F) imply specific increase of lipid storage.

Overexpression of ERK7 in the fat body (CG>ERK7) prominently (> 35%) reduced whole larval TAG levels (Fig 1G). This phenotype was dependent on ERK7 kinase activity and phosphorylation as expression of kinase-dead (K54R) and activation loop phosphorylation-deficient (T190A/Y192F) ERK7 mutants had no effect on TAG levels (Fig 1G). GFP-marked ERK7 overexpressing fat body clones (Fig 1H; marked by a yellow dotted line) displayed significantly reduced number and size of lipid droplets when compared to the neighboring control cells (marked by a white dotted line) (Figs 1I and J, and EV2C). The total lipid droplet area was significantly reduced by ERK7, even when normalized to cell size (Fig EV2D). These data demonstrate that ERK7 is sufficient to cell autonomously inhibit TAG storage. The fat body-specific role of ERK7 was further supported by the finding that ERK7 gene expression in the fat body is upregulated upon nutrient deprivation (Fig 1K). In contrast to fat body, expression of ERK7 in the insulin-producing cells (IPCs) did not lead to reduced TAG stores, ruling out the involvement of insulin signaling in the regulation of lipid metabolism in this setting (Fig EV2E).

To further assess the regulation of lipid metabolism by fat body ERK7 overexpression, we analyzed the fatty acid compositions of

---

**Figure 1.  ERK7 inhibits lipid storage in the fat body.**

A   Schematic presentation showing the generation of *ERK7*[1] mutants by CRISPR/Cas9 targeting.

B   *ERK7*[1] mutants have elevated triacylglycerol (TAG) levels (*N* = 4 replicates of ≥ 10 larvae/ replicate for each genotype).

C   Fat body-specific depletion of ERK7 by RNAi (BDSC 56939) leads to increased TAG levels (*N* = 4 replicates of 10 larvae/replicate for each genotype).

D   Representative immunofluorescent images of lipid droplet (LipidTOX) and nuclear (DAPI) staining in control and *ERK7*[1] mutant fat bodies of third instar larvae. Scale bar: 50 μm.

E   *ERK7*[1] mutant fat body cells contain more lipid droplets than control cells (*N* = 30 cells for each genotype).

F   After fed [13C]glucose, the *ERK7*[1] mutant fat bodies display elevated [13C]TAG/PE and unlabeled TAG/PE molar ratios (measured by mass spectrometry-based lipidomics, *N* = 3 replicates of 15 fat bodies/replicate, TAG = triacylglycerol, PE = phosphatidylethanolamine)

G   ERK7 overexpression in the fat body results in reduced TAG levels, while kinase-dead (K54R) and activation loop phosphorylation-deficient (T190A/Y192F) mutants of ERK7 do not influence the TAG storage (*N* = 4 replicates of ≥ 10 larvae/replicate for each genotype).

H–J   ERK7 expressing, GFP-marked, fat body clone (H, marked by yellow dotted line; scale bar: 50 μm) contains less (I) and smaller (J) lipid droplets than control cells (marked by white dotted line), visualized by LipidTOX staining (*N* = 11 for control and 4 for ERK7 overexpression).

K   qRT–PCR-based expression analysis of ERK7 from fat bodies of *w*[1118] larvae upon 6 h starvation. Expression of RP49 was used for normalization (*N* = 3 replicates of 10 fat bodies/replicate for each genotype).

Data information: *N* stands for the number of biological replicates. Error bars display standard deviation (SD). ns: not significant, **P* < 0.05, ***P* < 0.01, ****P* < 0.001 (Student's *t*-test).

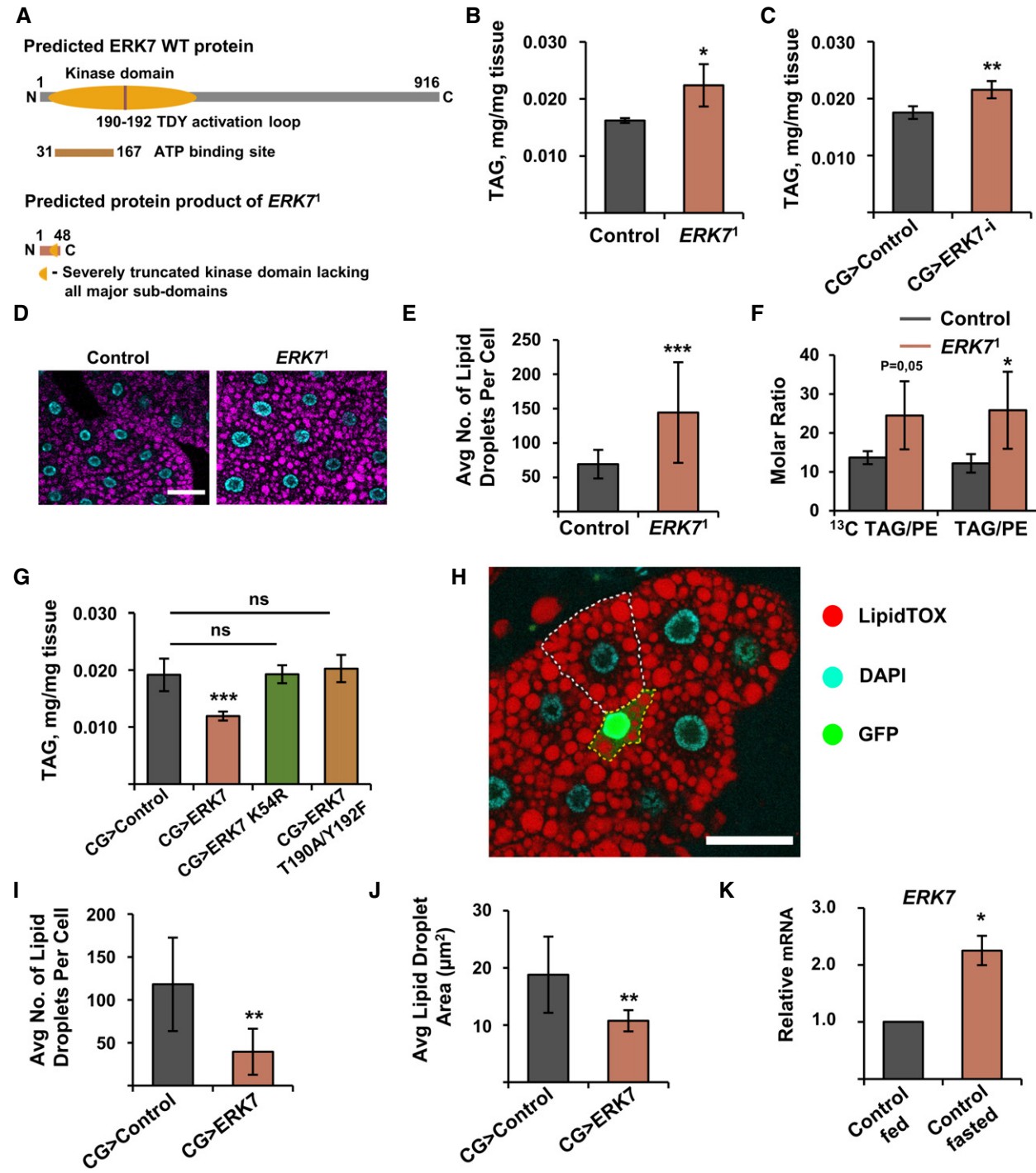

**Figure 1.**

the larval total lipids by gas chromatography–mass spectrometry. Fat body expression of ERK7 led to a significant reduction in the proportions of potentially *de novo* synthesized saturated fatty acids 14:0 and 16:0, which are direct products of fatty acid synthase (de Renobales & Blomquist, 1984), as well as monounsaturated fatty acids 16:1n-7 and 18:1n-9, which are formed during elongation and/ or desaturation reactions immediately after synthesis (Fig 2A and

B). In contrast, the proportions of 14:1n-7, 16:1n-9, and 14:1n-9 fatty acid isomers were elevated in CG>ERK7 larvae (Fig 2A and B). To the best of our knowledge, *Drosophila* tissues do not have delta-5 or delta-7 desaturase enzymes, which would be required to directly synthesize these isomers (Shen *et al*, 2010; Helmkampf *et al*, 2015). Chain shortening by peroxisomal beta oxidation of the delta-9 desaturated longer MUFAs, which is shown to be important

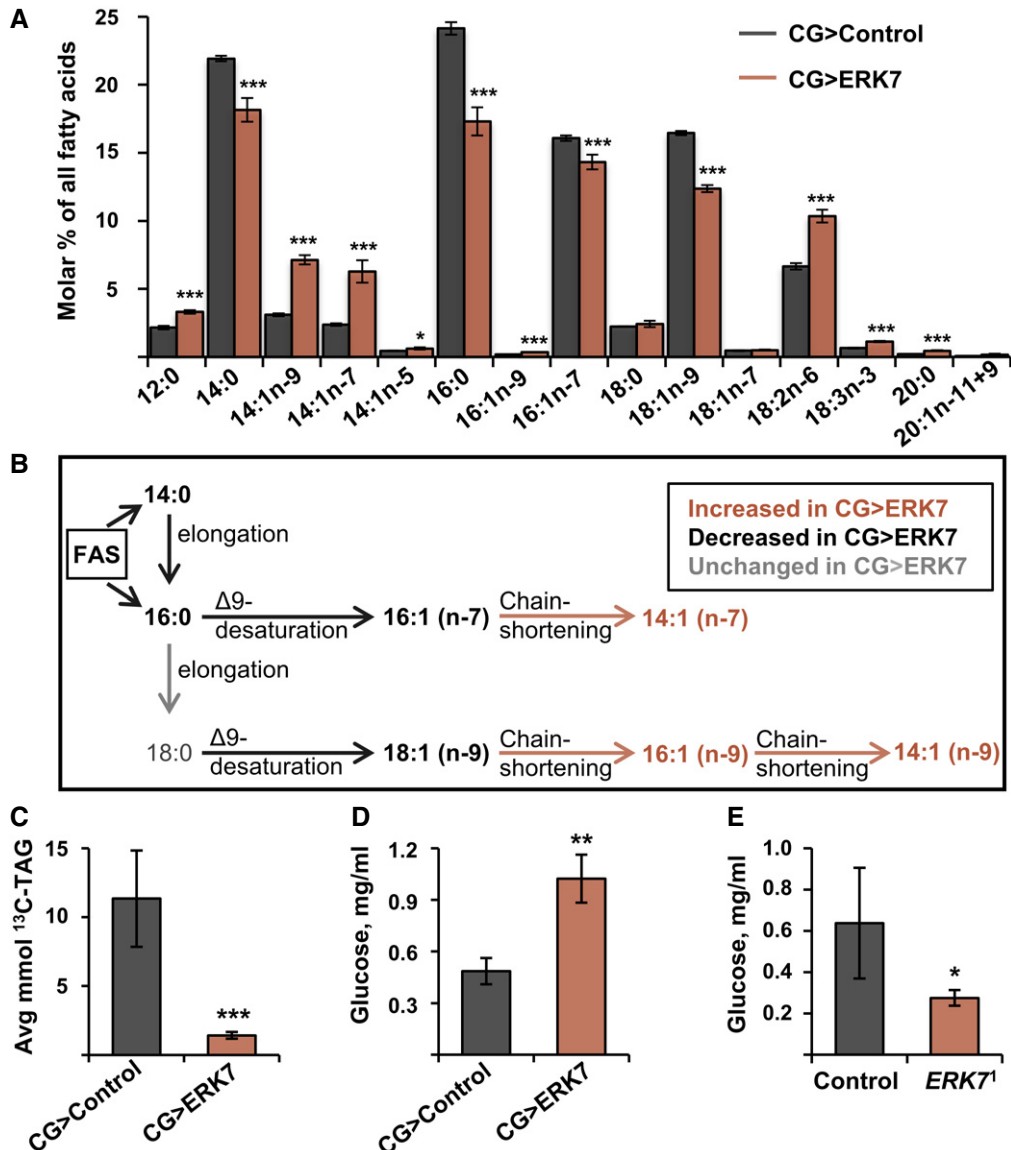

**Figure 2. ERK7 overexpression inhibits lipid biosynthesis.**

A   Fatty acid composition of total lipids in CG>control and CG>ERK7 larvae, measured by gas chromatography ($N = 3$ replicates of 15 larvae/replicate for each genotype).
B   Schematic presentation of biosynthesis of fatty acids and their likely chain shortening by peroxisomal partial β-oxidation.
C   Amount of glucose-derived [13C]-labeled carbons incorporated into the TAG pool of control and CG>ERK7 fat bodies ($N = 3$ replicates of 15 fat bodies/replicate for each genotype and diet).
D   Ectopic ERK7 expression in the fat body results in elevated hemolymph glucose levels ($N = 3$ replicates of 15 larvae/replicate for each genotype).
E   Circulating glucose levels are decreased in ERK7[1] mutants ($N = 4$ replicates of 15 larvae/replicate for each genotype).

Data information: $N$ stands for the number of biological replicates. Error bars display standard deviation (SD). *$P < 0.05$, **$P < 0.01$, ***$P < 0.001$ (Student's $t$-test).

for fatty acid metabolism in *Drosophila* (Faust *et al*, 2014), is the only documented process to produce the 14:1n-7, 16:1n-9, and 14:1n-9 fatty acid isomers. To directly analyze the activity of lipid biosynthesis, we fed the larvae with 13C-labeled glucose and analyzed the incorporation of the labeled carbons into the fat body TAG pool. Indeed, ERK7 fat body overexpression significantly reduced the glucose flux into fat body TAG stores (Fig 2C). Consistently, levels of circulating glucose were elevated in the CG>ERK7 larvae and reduced in *ERK7*[1] mutants (Fig 2D and E).

To get an unbiased view of ERK7-regulated downstream processes, we analyzed global gene expression changes in *ERK7*[1] mutants during feeding and acute nutrient deprivation. RNAseq analysis was performed in non-wandering 3rd instar control and *ERK7*[1] mutant larvae that were grown either on regular fly food or following full fasting for 6 h. For an overview of the impact of ERK7 on fasting-induced gene expression changes, the data were visualized on a four-way plot displaying comparisons of fed vs. fasted (control animals), as well as control vs. *ERK7*[1] mutant (fasted

conditions) (Fig 3A). This revealed an inverse correlation between genes responding to nutrient deprivation and to the loss of ERK7: 30 and 39% of fasting inhibited and activated genes, respectively, were less regulated in ERK7[1] mutants (Fig 3B). Gene set enrichment analysis revealed that ERK7 controls many gene groups involved in growth and metabolism (Fig 3C). Consistent with the phenotypic data, the analysis revealed significantly elevated expression of genes belonging to GO terms "lipid particle" and "regulation of lipid storage" in ERK7[1] mutants. A number of lipid synthesis and storage-related genes displayed attenuated downregulation upon fasting in ERK7[1] mutants (Fig 3D). Several of these genes, such as seipin, agpat3, ATPCL, and a lipid chaperone fabp were downregulated upon ERK7 fat body overexpression in fed animals (Fig 3E). Conversely, fasting-induced upregulation of lipases, such as CG34448 and Lip4, was attenuated in ERK7[1] mutants (Fig 3D). Expression of these genes was upregulated in the ERK7 overexpressing fat bodies of fed larvae (Fig 3F). Collectively, the ERK7-dependent gene expression changes are consistent with its observed role as a negative regulator of lipid storage.

### Fat body ERK7 inhibits growth tissue autonomously and systemically

In parallel to the metabolic phenotypes, we observed that ERK7 also regulates growth. ERK7[1] mutant larvae displayed accelerated larval growth rate, as evidenced by increased body weight at 72 h after egg deposition (Fig 4A) and earlier pupariation compared to the control larvae (Fig 4B). No changes in the pupal volume were observed (Fig EV3A). When animals are fed on a limited nutrient source, wild-type animals do not grow further in order to preserve their energy. In contrast, ERK7[1] mutants failed to cease their growth upon 24 h of fasting (Fig 4C), which coincided with reduced survival of ERK7[1] mutant larvae upon fasting (Fig 4D). Fat body-specific depletion of ERK7 by two independent RNAi lines also led to a moderate increase in the larval growth rate (Figs 4E and F, and EV3B), but had no impact on pupal volume (Fig EV3C), phenocopying ERK7[1] mutants. Fat body-specific overexpression of ERK7 (CG>ERK7) led to developmental delay (Fig 4G and H) and substantial (> 20%) reduction of pupal volume (Fig 4I). ERK7 expression in the fat body inhibited growth to a similar extent as the expression in the IPCs (Fig EV3D). This growth inhibition required ERK7 kinase activity and phosphorylation, as evidenced by the observation that expression of kinase-dead (K54R) and activation loop phosphorylation-deficient (T190A/Y192F) mutants of ERK7 in the fat body did not influence growth (Fig 4H and I).

Consistent with the increased growth rate, ERK7[1] mutant fat body cells displayed increased nuclear size (Fig 5A and B) as well as increased relative size of the nucleolus (Fig 5C). To explore if ERK7 is sufficient to inhibit growth in a cell autonomous manner, we induced GFP-marked fat body clones with ERK7 overexpression. The nuclei of the ERK7-expressing cells were significantly smaller in size (Fig 5D and E). The ERK7-expressing clones also displayed a reduced relative size of the nucleolus (Fig 5F). In line with the impaired growth and reduced nucleolar size, quantitative RT–PCR analysis showed inhibited expression of fat body ribosomal RNA products (rRNA) of RNA polymerases I (Fig 5G) and III (Fig 5H) upon ERK7 fat body overexpression.

### PWP1 is a target of ERK7 involved in lipid homeostasis and growth control

Our findings on the role of ERK7 in Pol I- and Pol III-mediated rRNA expression led us to test the possible involvement of chromatin-binding protein PWP1, which we have recently identified as a positive regulator of Pol I and Pol III (Liu et al, 2017; Liu et al, 2018). PWP1 is activated by Insulin/mTOR-mediated phosphorylation, which can be visualized by using a Phos-tag gel (Liu et al, 2017). Intriguingly, ERK7 co-expression with PWP1 in S2 cells induced an additional band in the PWP1 Phos-tag blot (Fig 6A, red arrow), while the Insulin/mTOR-inducible PWP1 phosphorylation pattern (Fig 6A, dark blue arrows) remained unchanged upon ERK7 expression. This implies that PWP1 is phosphorylated in an ERK7-dependent manner, either directly or indirectly. To test if PWP1 is regulated by ERK7 in vivo, we analyzed PWP1 localization, which is known to be regulated by nutrient-responsive cues (Liu et al, 2017). Interestingly, PWP1 nucleolar localization was significantly increased in the ERK7[1] mutant fat bodies (Fig 6B and C). ERK7-overexpression in the fat body also had a significant effect on PWP1 subcellular localization. ERK7 expressing clones exhibited significantly reduced nuclear to cytoplasmic ratio compared to control cells (Fig 6D and E). Thus, ERK7 is necessary and sufficient to regulate PWP1 subcellular localization in the fat body.

While PWP1 is known to regulate growth (Liu et al, 2017), its possible role in lipid homeostasis has remained unexplored. We observed that fat body-specific knockdown of PWP1 by CG-GAL4 driver caused a significant reduction in TAG levels (Fig 6F), phenocopying ERK7 overexpression. Moreover, genetic epistasis experiment revealed that fat body-specific knockdown of PWP1 strongly

---

**Figure 3. ERK7 regulates lipid metabolism gene expression.**

A  Four-way plot (t-statistics) presenting the ERK7-dependent fasting-responsive genes (adj.P.val < 0.05). Dark blue indicates significant in both comparisons; dark turquoise, significant in one comparison; light turquoise, not significant.

B  Venn diagrams of the fasting-downregulated/ERK7-dependent transcriptome (adj.P.val < 0.05) and fasting upregulated/ERK7-dependent transcriptome (adj.P.val < 0.05).

C  Bar plots of selected ERK7-regulated pathways with overrepresentation of significantly changing genes (adj.P.val < 0.05, LFC < −0.5 or LFC > 0.5). The bar plot shows enrichment (−log 10 adj.P.val, generated by R/Bioconductor piano package). Blue color denotes downregulated in mutants and red denotes upregulated in mutants. G—Gene Ontology (GO), K—KEGG Pathway, R—Reactome Pathway Database.

D  Heatmaps displaying expression patterns of selected genes involved in lipid anabolism and lipid catabolism. Color key displays scaled $\log_2$ gene expression.

E, F  Expression analysis of Seipin, Agpat3, ATPCL, fabp, Lip4, and CG34448 by qRT–PCR. RP49 was used for normalization (N = 3 replicates of ≥10 larvae/replicate for each genotype).

Data information: N stands for the number of biological replicates. Error bars display standard deviation (SD). *P < 0.05, **P < 0.01, ***P < 0.001 (Student's t-test).

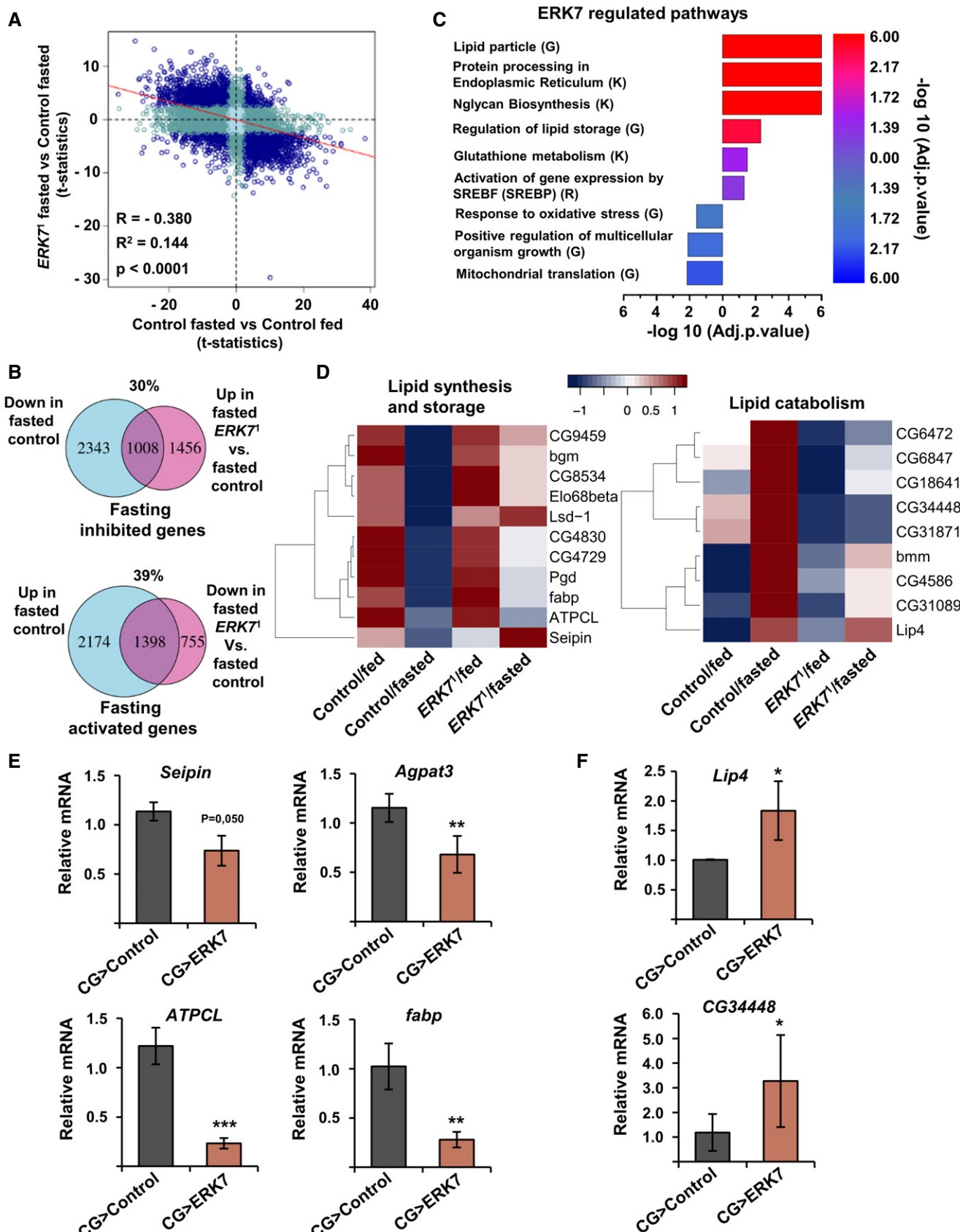

**Figure 3.**

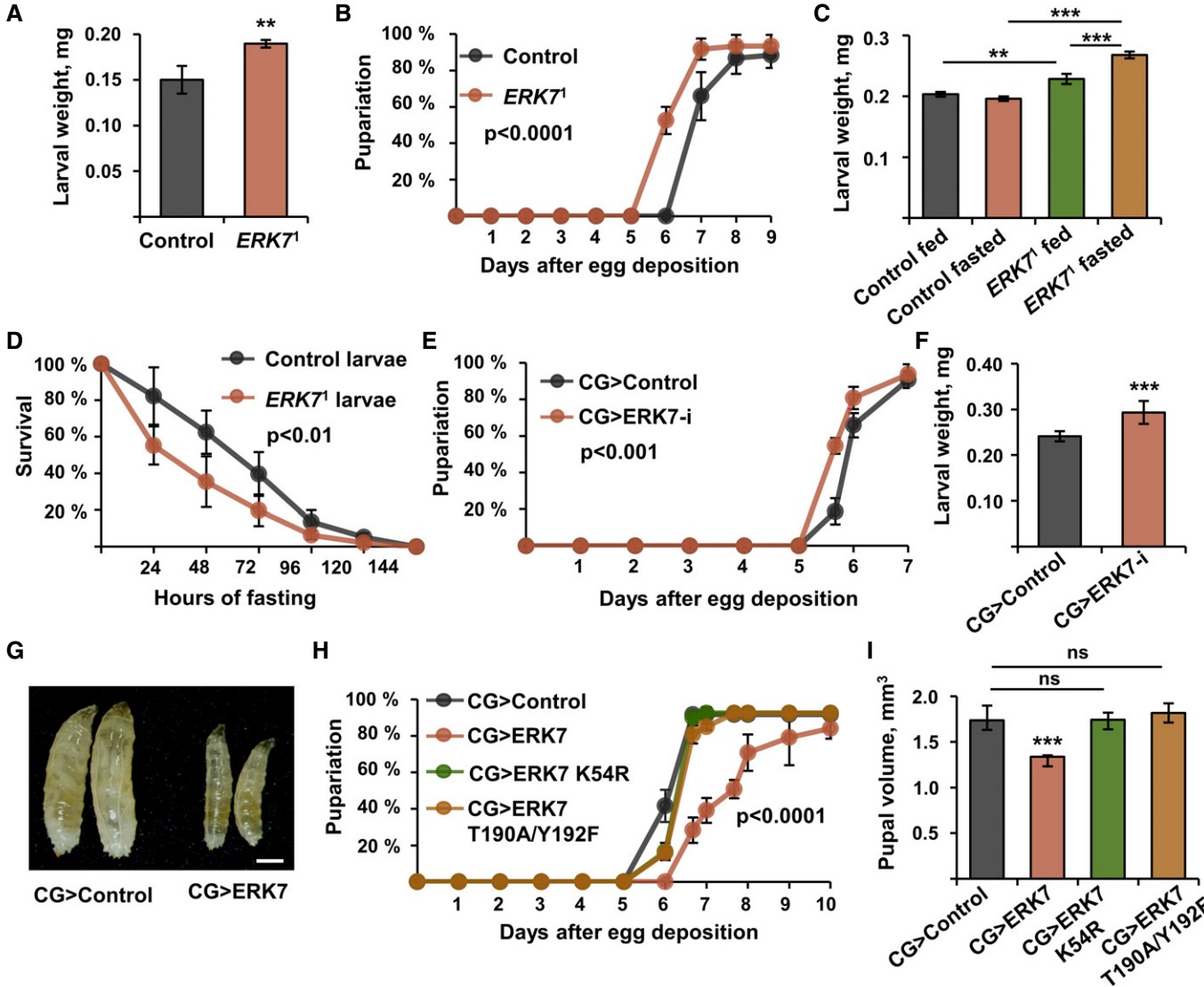

**Figure 4. Fat body ERK7 inhibits cell autonomous and systemic growth.**

A  Weight of control and *ERK7*[1] mutant larvae at 72 h after egg deposition (AED) ($N = 4$ replicates of 12 larvae/replicate for each genotype).

B  Pupariation kinetics of control and *ERK7*[1] mutants ($N = 4$ replicates of 30 larvae/replicate for each genotype). $P < 0.0001$, analyzed by Log-rank test.

C  *ERK7*[1] mutants fail to suppress growth upon 24 h of starvation ($N \geq 4$ replicates of 15 larvae/replicate for each genotype).

D  *ERK7*[1] mutant larvae ($N = 4$ replicates of 24 larvae/replicate for each genotype, $P < 0.01$ analyzed by Log-rank test) are sensitive to nutrient deprivation.

E, F  Fat body-specific knockdown of ERK7 by RNAi (BDSC 56939) leads to a moderately accelerated pupariation rate (E, $P < 0.001$ analyzed by Log-rank test, $N = 8$ replicates of 30 larvae/replicate for each genotype) and an increase in larval weight at 72 h AED (F, $N = 5$ replicates of 15 larvae/replicate for each genotype).

G  A representative image showing that fat body-specific overexpression of ERK7 by the *CG-GAL4* driver (CG>ERK7) impairs larval growth. Scale bar: 0.5 mm.

H  Ectopic expression of ERK7 leads to delayed pupariation ($P < 0.0001$, analyzed by Log-rank test), while kinase-dead (K54R) and activation loop phosphorylation-deficient (T190A/Y192F) mutants of ERK7 do not influence pupariation rate ($N = 4$ replicates of 30 larvae/replicate for each genotype).

I  In contrast to ERK7 wt, kinase-dead (K54R) and activation loop phosphorylation-deficient (T190A/Y192F) mutants of ERK7 do not influence the pupal volume ($N = 4$ replicates of 10 pupae/replicate for each genotype).

Data information: $N$ stands for the number of biological replicates. Error bars display standard deviation (SD). **$P < 0.01$, ***$P < 0.001$, ns—not significant (Student's *t*-test).

suppressed the increase of lipid droplet numbers caused by ERK7 knockdown (Fig 6G and H). Loss of PWP1 also suppressed the increase of nuclear size by ERK7 RNAi (Fig 6I). These data are consistent with the conclusion that PWP1 contributes to lipid storage and growth downstream of ERK7.

**ERK7 controls lipid storage and growth through the transcription factor Sugarbabe**

Earlier studies have shown the role of Gli-similar 2 transcription factor ortholog, Sugarbabe, in promoting lipid anabolism and

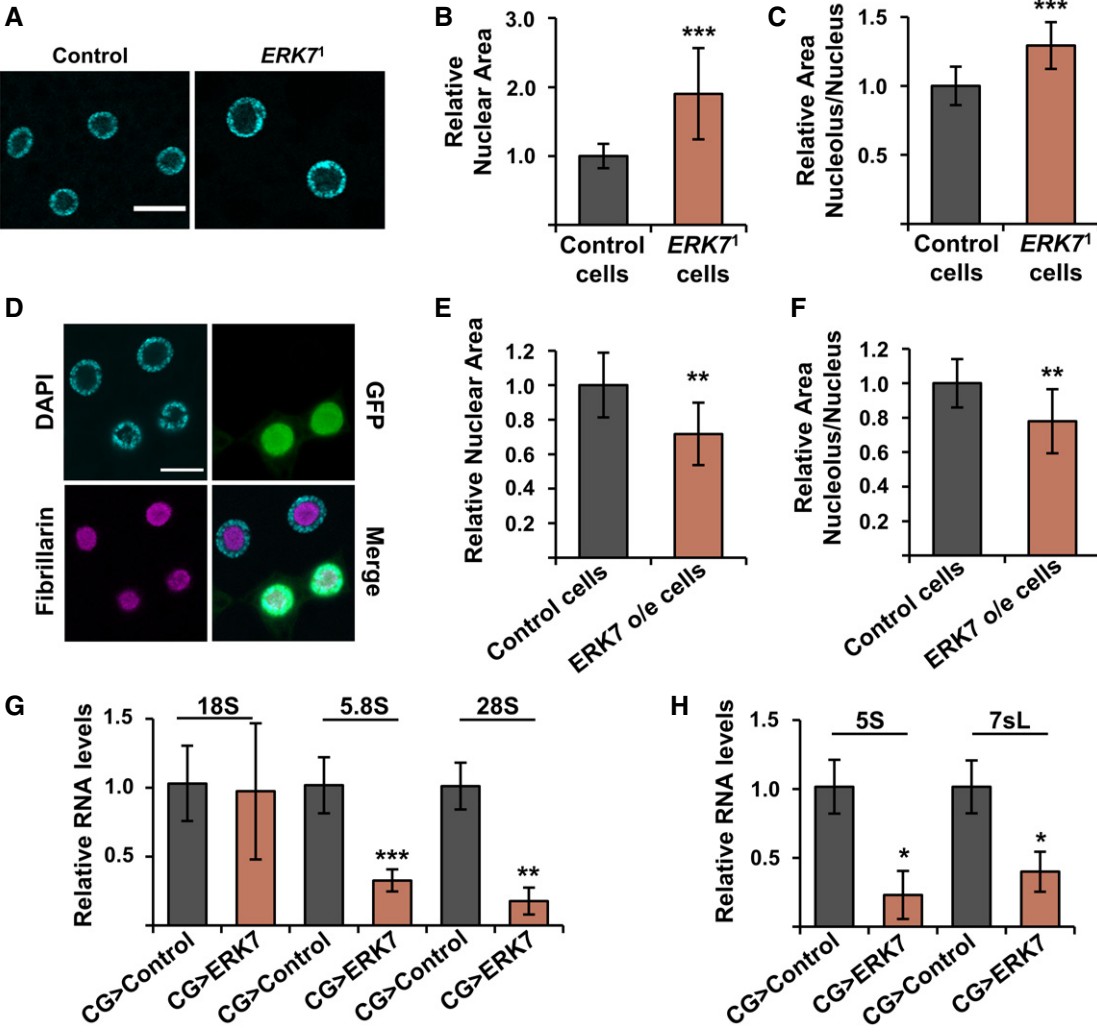

**Figure 5. ERK7 regulates nucleolar size and ribosome biogenesis.**

A   Representative immunofluorescent images of control and *ERK7*[1] mutant fat bodies with DAPI staining to visualize nucleus. Scale bar: 30 μm.

B, C   *ERK7*[1] mutant fat bodies display increased nuclear area (B, *N* > 160 cells, obtained from 5 independent fat bodies) and increased nucleolar/nuclear area ratio (C, *N* > 80 cells for nucleolus/nucleus ratio, obtained from 5 independent fat bodies), when compared to control fat bodies.

D–F   ERK7 expressing fat body clones (D, marked by GFP, scale bar: 20 μm) display reduced nuclear area (E) and reduced nucleolar/nuclear area ratio (F), when compared to control cells. Nuclei were visualized by DAPI staining, nucleoli were visualized by anti-fibrillarin antibodies (*N* = 10).

G, H   Expression analysis of Pol I (G) and Pol III (H) target RNAs by qRT–PCR. Expression of CDK7 was used for normalization (*N* = 3 replicates of 10 fat bodies/replicate for each genotype).

Data information: *N* stands for the number of biological replicates. Error bars display standard deviation (SD). *$P < 0.05$, **$P < 0.01$, ***$P < 0.001$, ns—not significant (Student's *t*-test).

---

inhibiting lipid catabolism (Zinke *et al*, 2002; Mattila *et al*, 2015; Luis *et al*, 2016). Interestingly, *sugarbabe* expression was significantly downregulated in the fat body upon PWP1 depletion (Fig 7A). Consistently with PWP1 inhibition by ERK7, *sugarbabe* expression was also strongly downregulated by ectopic fat body expression of ERK7 (Fig 7B). Moreover, *sugarbabe* expression was elevated in *ERK7*[1] mutants (Fig 7C). Genetic epistasis experiments were performed to test whether changes in Sugarbabe levels contribute to the phenotypes caused by ERK7. Hypomorphic *sugarbabe* mutant *in trans* with a deficiency (*sug*[17Δ]/Df(2R)Exel7123) suppressed the elevated lipid droplet numbers in *ERK7*[1] mutants

(Fig 7D and E). Moreover, Sugarbabe overexpression partially rescued the TAG levels in the CG>ERK7 larvae (Fig 7F). The observed rescue was not due to impaired *ERK7* expression in the double transgenic lines (Fig EV4A and B). Thus, Sugarbabe acts downstream of ERK7 to control lipid metabolism.

Interestingly, the *sugarbabe* mutant also suppressed the elevated nuclear size in *ERK7*[1] mutants (Fig 7G). Moreover, Sugarbabe overexpression significantly suppressed the developmental delay (Fig 7H) and pupal volume phenotypes caused by ERK7 overexpression (Fig 7I). *Sugarbabe* expression is strongly elevated upon sugar feeding (Zinke *et al*, 2002; Mattila *et al*, 2015). Therefore, we tested

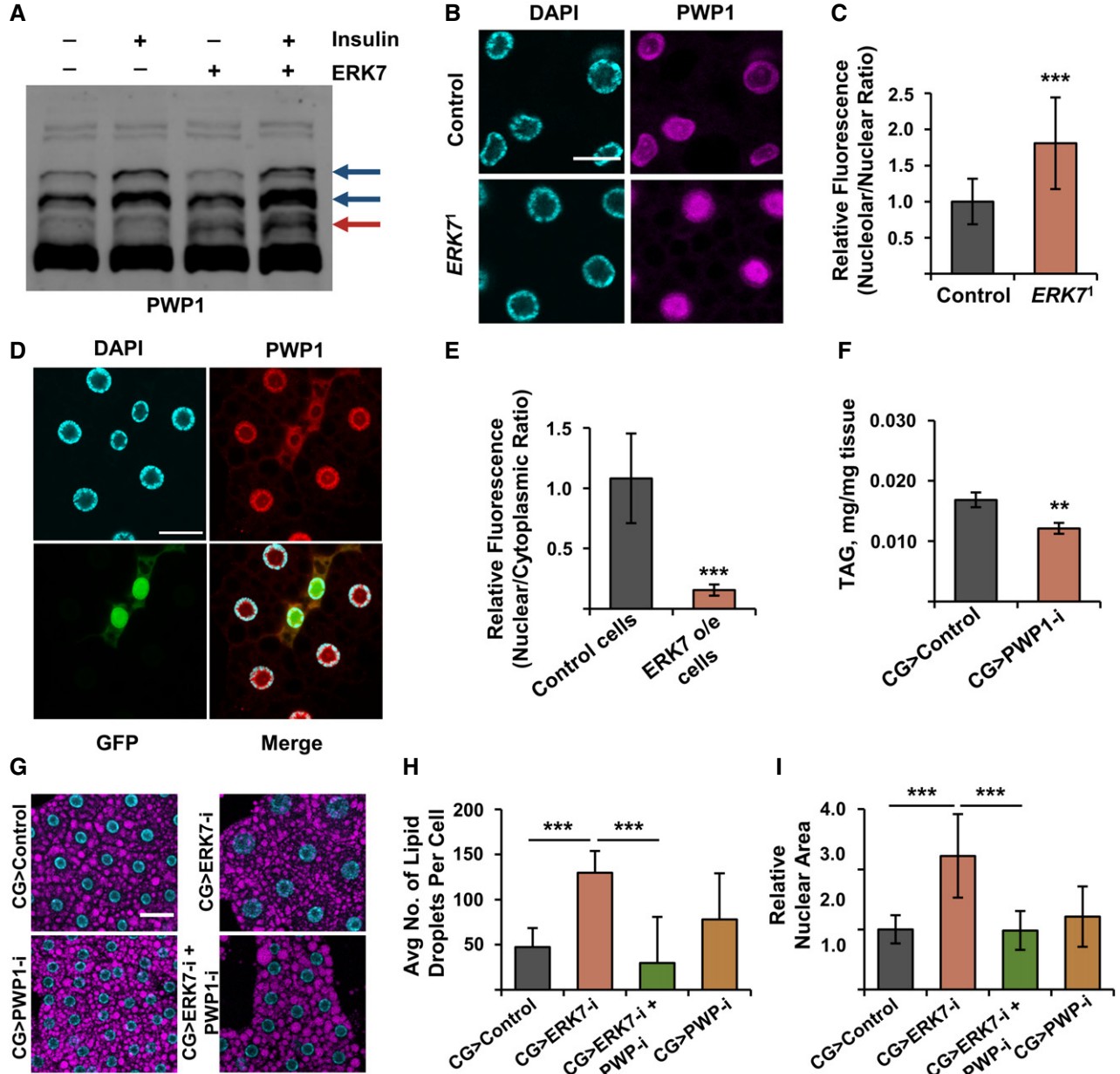

**Figure 6. PWP1 acts downstream of ERK7 to control growth and lipid storage.**

A    ERK7 overexpression leads to phosphorylation of PWP1, as shown by appearance of an extra band on Phos-tag Western blot (marked with red arrow). Insulin-inducible bands (dark blue arrows) remain unresponsive to ERK7.

B    Representative immunofluorescent images of PWP1 localization in fat bodies of *ERK7*[1] mutant and control third instar larvae. Scale bar: 20 μm.

C    PWP1 localization displays increased nucleolar/nuclear ratio in *ERK7*[1] mutant fat bodies compared to control (*N* = 20 cells from 5 independent fat bodies per genotype).

D    Representative immunofluorescent images of PWP1 localization in fat bodies of third instar larvae. ERK7 overexpression (GFP-marked clones) alters the subcellular localization of PWP1. Scale bar: 30 μm.

E    PWP1 localization displays reduced nuclear/cytoplasmic ratio in ERK7 overexpressing clones, compared to control cells (*N* = 10 cells from at least 8 different fat bodies per genotype).

F    Fat body-specific depletion of PWP1 by RNAi (NIG-Fly 6751R-3) leads to decreased TAG levels (*N* = 4 replicates of ≥ 10 larvae/replicate for each genotype).

G    Representative immunofluorescent images of fat bodies in a genetic epistasis experiment using ERK7 and PWP1 fat body (CG-GAL4) knockdown with LipidTOX staining and DAPI to visualize lipid droplets and nucleus, respectively. Scale bar: 50 μm.

H, I    Fat body-specific knockdown of PWP1 (NIG-Fly 6751R-1) suppresses increased lipid droplet number (H; *N* > 30 cells per genotype) and increased nuclear area (I; *N* > 45 cells per genotype) caused by ERK7 knockdown (BDSC 56939).

Data information: *N* stands for the number of biological replicates. Error bars display standard deviation (SD). **P < 0.01, ***P < 0.001 (Student's *t*-test).

if the growth impairment of fat body-specific ERK7 expressing larvae could be rescued by dietary sugar. CG>ERK7 larvae survived very poorly on 20% yeast food, displaying only 30% pupariation rate and prominent developmental delay (Fig 7J). Strikingly, the addition of 20% sugar to the 20% yeast food significantly improved survival and developmental rates in the CG>ERK7 larvae (Fig 7J). We confirmed that sugar feeding led to a significant increase in *sugarbabe* expression levels in CG>ERK7 larvae (Fig EV4C). Thus, Sugarbabe also contributes to growth control downstream of ERK7, further highlighting the coordinated control of growth and lipid storage by ERK7.

## Discussion

Our study uncovered a new regulatory mechanism of lipid metabolism and growth *in vivo*. We report that an atypical MAP kinase ERK7 inhibits lipid storage and cell growth in the *Drosophila* fat body. ERK7 expression is upregulated by fasting and it contributes to the physiological adaptation to nutrient shortage. Mechanistically, ERK7 regulates the subcellular localization of the chromatin-binding protein PWP1 and inhibits the expression of the transcription factor, Sugarbabe. By regulating these downstream effectors, ERK7 counteracts the output of anabolic nutrient sensors mTORC1 and Mondo-Mlx.

While several functions have been assigned to ERK7 in cultured cells (Iavarone *et al*, 2006; Groehler & Lannigan, 2010; Xu *et al*, 2010; Cerone *et al*, 2011; Zacharogianni *et al*, 2011; Colecchia *et al*, 2012; Chia *et al*, 2014; Colecchia *et al*, 2015; Jin *et al*, 2015; Rossi *et al*, 2016; Colecchia *et al*, 2018), our understanding of its physiological role has remained limited. We discovered that, in addition to inhibitory role in the IPCs (Hasygar & Hietakangas, 2014), ERK7 has an important regulatory role in the fat body, where it inhibited both cell autonomous and organismal growth. Fat body ERK7 was sufficient to inhibit rRNA expression, which is a well-established mechanism for nutrient-dependent growth control. Notably, inhibition of ribosome assembly in the IPCs (e.g., by inhibition of Rio kinases) led to *ERK7* upregulation (Hasygar & Hietakangas, 2014), implying that ERK7 may have a role as a feedback regulator of ribosome biogenesis. Human ortholog of ERK7, MAPK15/ERK8, was earlier shown to promote autophagy (Colecchia *et al*, 2012), which together with ribosome biogenesis, controls cellular amino acid

balance. As mTOR signaling inhibits autophagy and activates ribosome biogenesis (Scott *et al*, 2004; Grewal *et al*, 2007), ERK7 appears to counteract these key downstream effects of the mTOR pathway. A recently established regulator of ribosome biogenesis and growth, PWP1, was identified as a downstream target for ERK7 signaling. ERK7 expression induced phosphorylation of PWP1 detected by Phos-tag gel, while having no influence on the Insulin/mTORC1-dependent bands. This suggests that ERK7 controls PWP1 in parallel, rather than upstream, of mTORC1. ERK7 activity in the fat body cells caused a prominent change in PWP1 subcellular localization. While loss of ERK7 increased PWP1 localization in the nucleolus, ERK7 overexpression led to increased cytoplasmic localization. We have previously shown that nucleolar PWP1 is lost upon inhibition of mTOR (Liu *et al*, 2017). Our present observations of the regulation of PWP1 phosphorylation and subcellular localization by ERK7 further highlight the role of PWP1 as a downstream effector of nutrient-responsive signaling pathways.

The human ortholog of *ERK7*, *MAPK15*, is associated with obesity (Li *et al*, 2012). However, the functional role of ERK7 in energy metabolism of animals has not been previously addressed. Here, we observed that ERK7 has an antiadipogenic role in *Drosophila*. Several lines of evidence imply that ERK7 inhibits lipid metabolism and storage tissue autonomously in the fat body. Firstly, ERK7 overexpression in the fat body clones results in smaller and fewer lipid droplets, while *ERK7* mutant fat bodies possess elevated number of lipid droplets *per* cell. Secondly, fat body-specific expression of ERK7 inhibits the flux of glucose-derived carbon into the TAG pool. Thirdly, in a genome-wide expression analysis, *ERK7* mutants displayed deregulated expression of genes involved in lipid synthesis, storage, and catabolism. Our genetic epistasis experiments further revealed that the lipid storage phenotypes are dependent on the Gli-similar transcription factor Sugarbabe, whose expression is inhibited by ERK7. We found that *sugarbabe* expression was downregulated by the inhibition of PWP1, a downstream target of ERK7. Consistently, PWP1 was necessary for lipid storage downstream of ERK7, thus broadening the role of PWP1 as an anabolic regulator *in vivo*. Previous studies have shown that *sugarbabe* is one of the most strongly sugar-responsive genes in *Drosophila* (Zinke *et al*, 2002; Mattila *et al*, 2015). It is a direct target of the sugar-sensing transcription factor complex Mondo-Mlx and its expression is also promoted by the Dawdle-induced TGF-beta/Activin signaling pathway, which signals through the transcription

**Figure 7. Sugarbabe is a downstream effector of ERK7 and PWP1.**

A, B  Expression of *sugarbabe* is downregulated in fat bodies upon knockdown of PWP1 (A) and ectopic expression of ERK7 (B) (N = 3 replicates of 10 fat bodies/replicate for each genotype).

C  *ERK7*[1] mutants fail to maximally downregulate *sugarbabe* expression upon fasting (N = 4 replicates of 10 larvae/replicate for each genotype).

D  Representative immunofluorescent images of fat bodies in a genetic epistasis experiment using *ERK7*[1] and *sug* mutants (*sug*[17Δ]/Df(2R)Exel7123) with LipidTOX and DAPI staining to visualize lipid droplets and nucleus, respectively. Scale bar: 50 μm.

E  *sug* mutant (*sug*[17Δ]/Df(2R)Exel7123) suppresses increased lipid droplet number observed in *ERK7*[1] mutants (N > 20).

F  Ectopic *sugarbabe* expression in the fat body partially rescues TAG levels in the CG>ERK7 larvae (N = 4 replicates of ≥10 larvae/replicate for each genotype).

G  *sug* mutant (*sug*[17Δ]/Df(2R)Exel7123) suppresses increased nuclear size phenotype of *ERK7*[1] mutant fat body cells (N ≥ 25 cells per genotype).

H, I  Ectopic Sugarbabe expression in the fat body partially restores the pupariation rate (H) (P < 0.001 for the rescue, analyzed by Log-rank test, N = 4 replicates of 30 larvae/replicate for each genotype) and pupal volume of CG>ERK7 larvae (I) (N = 4 replicates of 10 pupae/replicate for each genotype).

J  Addition of 20% sucrose improves the growth rate and survival of CG>ERK7 larvae. Pupariation kinetics of CG>ERK7 larvae on 20% yeast or 20% yeast + 20% sucrose diet (P < 0.0001 for the rescue, analyzed by Log-rank test, N = 4 replicates of 30 larvae/replicate for each genotype and diet).

Data information: N stands for the number of biological replicates. Error bars display standard deviation (SD). *P < 0.05, **P < 0.01, ***P < 0.001 (Student's *t*-test). Expression of RP49 was used for normalization of (A–C).

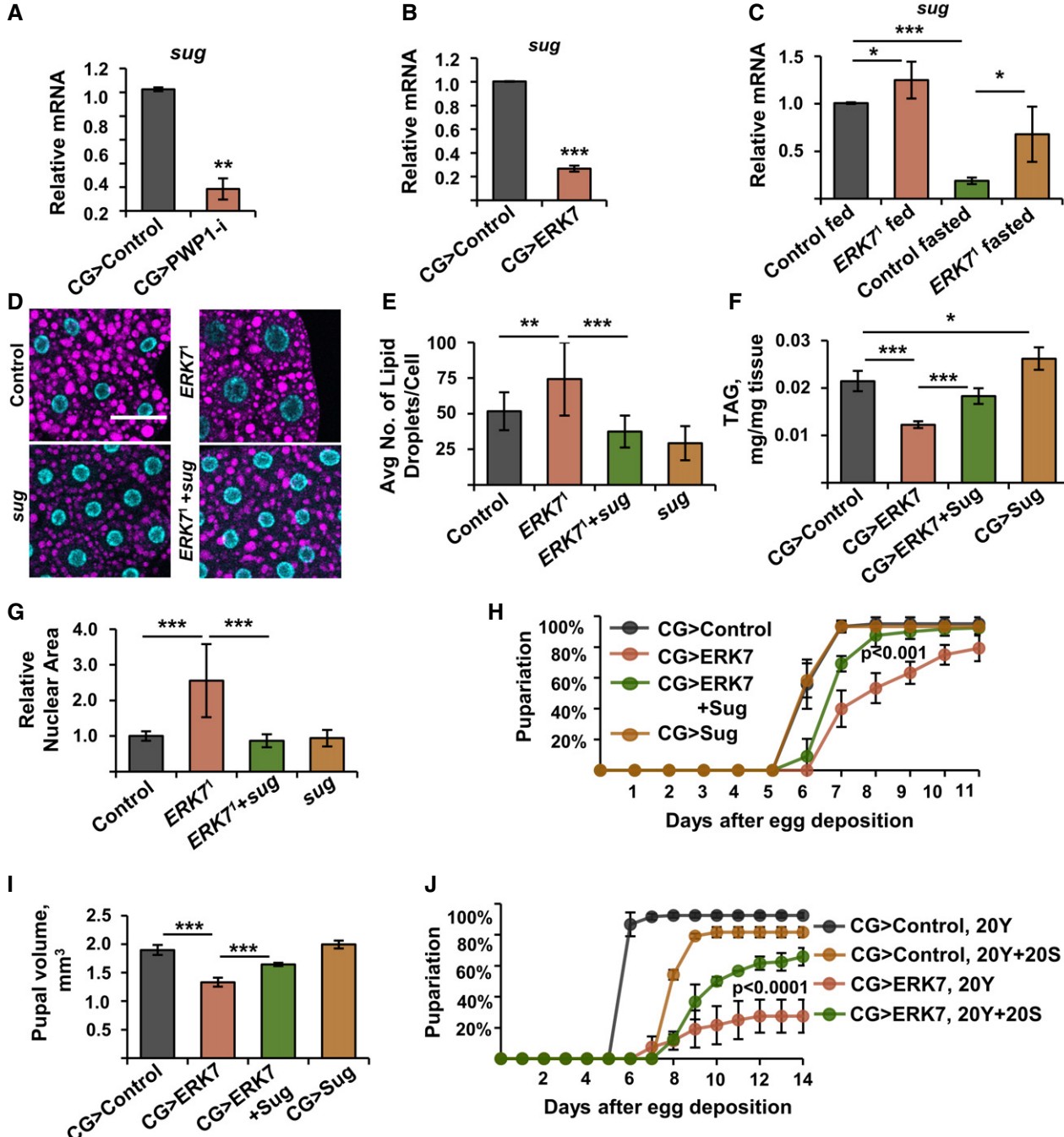

**Figure 7.**

factor SMAD2 (Mattila *et al*, 2015). Thus, Sugarbabe regulation converges several nutrient-dependent inputs to control lipid metabolism. Notably, we also discovered that Sugarbabe was necessary and sufficient to control growth downstream of ERK7, both at the level of individual cells as well as the whole organism. This highlights that the regulation of fat body lipid homeostasis is closely coupled to growth control.

Wild *Drosophila melanogaster* larvae face competition against their peers and the micro-organisms fermenting their nutrient source. Therefore, there is a selective advantage to be gained by fast

larval development without compromising body size, such as that observed in the *ERK7* mutants. However, such rapid growth on a rich diet comes with a cost, as tolerance to fasting was compromised in the *ERK7* mutants. Recent studies have shown macronutrient-dependent differences in larval performance between closely related *Drosophila* species. For example, *D. simulans* larvae develop significantly better on a high sugar diet than its specialist relative *D. sechellia*, which outperforms *D. simulans* on limited nutrition (Melvin *et al*, 2018). Similarly, *D. biarmipes*, which feeds on rotting fruit with protein-rich micro-organisms, shows optimal larval performance at

higher protein levels than *D. suzukii*, which colonizes fruit that is still ripe (Silva-Soares *et al*, 2017). It will be interesting to explore whether natural variation in the *ERK7* gene contributes to animal adaptation to habitats with distinct nutrient availabilities.

# Materials and Methods

### Fly stocks and husbandry

Flies were maintained at 25°C in a 12-h light/12-h dark cycle, on standard *Drosophila* medium (agar 0.6% (*w/v*), semolina 3.2% (*w/v*), malt 6.5% (*w/v*), dry baker's yeast 1.8% (*w/v*), propionic acid 0.7% (*v/v*), and Nipagin (methylparaben) 2.4% (*v/v*). Driver lines used: dILP2-Gal4, UAS-GFP (Rulifson *et al*, 2002), CG-Gal4 (Hennig *et al*, 2006). $w^{1118}$ crossed to the respective Gal4-lines were used as controls. Stocks used included: UAS-ERK7 (Hasygar & Hietakangas, 2014), UAS-ERK7-RNAi (BDSC 56939, VDRC v109661), UAS-Sug (FlyORF F000721), UAS-PWP1-RNAi (NIG-Fly 6751R-3, NIG-Fly 6751R-1), and Actin<CD2<GAL4 GFPnls. To generate mosaic clones, larvae (genotype*: HsFLP/+; UAS-ERK7/+; Actin<CD2<Gal4 GFPnls/+*) were heat-shocked at 37°C for 60 min at 24 h after egg laying. $Sug^{17\Delta}$ mutant was used in *trans* with a deficiency Df(2R)Exel7123 as described in (Mattila *et al*, 2015).

Generation of kinase defective ERK7 overexpression flies: ERK7 K54R and ERK7 T190A/Y192F cDNA were amplified by PCR from pMT-ERK7K54R-V5-His and pMT-ERK7T190A/Y192F-V5-His (Zacharogianni *et al*, 2011), cloned into the pUAST-attB vector and confirmed by sequencing. Both the transgenes were directed to the attP landing site at 22A2, similar to the wt ERK7. Transgenic flies were constructed by BestGene Inc.

### CRISPR/Cas9-mediated mutagenesis of the *ERK7* gene

*ERK7* mutant flies were generated using the CRISPR/Cas9 (Kondo & Ueda, 2013). The following sgRNA sequence was used: TGCCTATGG-CATCGTCTGGAAGG. Embryo injections were carried out by Bestgene Inc. Progeny was screened for the mutations using T7 Endonuclease I (NEB) assay (Vouillot *et al*, 2015). One mutant line was identified ($ERK7^1$) which was further confirmed by direct sequencing of the target region. $ERK7^1$ line was found to contain a deletion of 10 nucleotides (CDS: 111 – 120, CATCGTCTGG). The deletion also resulted in a frameshift and introduced a premature stop codon after 48 amino acids. This deletion disrupted the open reading frame and led to a severely truncated polypeptide (48 AA) lacking the predicted ATP binding site as well as the activation loop (Abe *et al*, 1999; Marchler-Bauer & Bryant, 2004). The first alternate in-frame ATG sequence downstream of the mutated site encodes methionine 107. Should that codon be utilized as a start codon in the mutant, the resulting polypeptide would lack major parts of the kinase domain, including the predicted active site, ATP binding site, and polypeptide substrate binding site. Thereby, we conclude that $ERK7^1$ is a likely null allele. Another line, which underwent the same screening process, but did not contain any mutations in the *ERK7* gene, was used as control. $ERK7^1$ and control lines were backcrossed into $w^{1118}$ genetic background (VDRC 60000) for five generations and maintained as balanced stocks.

### Fertility assay

Ten male flies from control or $ERK7^1$ lines were crossed with ten $w^{1118}$ females in four replicates and were allowed to lay eggs for 3 days on standard laboratory food. Total number of adults hatched was counted after 18 days.

### Fasting conditions

Larvae: Equal number (30/vial) of freshly hatched L1 larvae was grown on standard laboratory food for 2 days. Early 3rd instar larvae (72 h AED, pre-critical weight) were transferred to 48-well plates (1 larva/well) containing PBS + 0.5% agar + 0.2% Erioglaucine dye (Dr. Oetker). The number of surviving larvae was counted every 24 h.

### RNA sequencing

Freshly hatched L1 larvae were grown at equal density (30/vial) on standard *Drosophila* medium. Third instar larvae (90 h AED) were transferred to starvation medium (PBS + 0.5% agarose) for 6 h. Total RNA was isolated from both fed and starved male larvae (control and $ERK7^1$) using Nucleospin RNA II kit (Macherey-Nagel). Transcriptome sequencing (RNAseq) was performed with Illumina NextSeq500 technology to an average depth of 20 M clean reads per sample.

### Bioinformatic analysis of RNAseq data

FastQC v.0.11.2 was used for quality assessment and Trimmomatic v.0.33 for trimming the data. The minimum read length of 36 bases was required. Minimum read quality score of 20 was used. The reads were scanned with 4-base sliding window and trimmed when the average quality per base dropped below 15. The reads were mapped with TopHat (v.2.1.0) to the *D. melanogaster* reference genome (FlyBase R6.10). Transcription was quantified strand-specifically with HTSeq (v.2.7.6) on the level of annotated exons (Flybase) with reads below quality score of 10 discarded. The differential expression among different samples and conditions was quantified with limma package (v.3.28.8) implemented in R/Bioconductor with Benjamini–Hochberg correction. The genes with very low counts (cpm < 1 in more than one replicate per condition) were filtered.

### Gene set enrichment analysis

Gene set enrichment analysis was performed with R/Bioconductor package piano (v.1.12.1) for significant genes (Control starved vs. $ERK7^1$ mutants starved, adj.P.val < 0.05) with *t*-statistics by using gsa with gsea algorithm and row sampling (1,000 permutations). The manually downloaded databases consisted of KEGG, Reactome, Wikipathways, and GO.

### RNA extraction and quantitative RT–PCR

RNA extraction from larval fat bodies: 10 fat bodies from stage-matched third instar non-wandering larvae were dissected and RNA was extracted using Nucleospin RNA XS kit (Macherey-Nagel) according to the manufacturer's protocol.

RNA extraction from whole larvae: Five stage-matched third instar non-wandering larvae were homogenized and RNA was extracted using Nucleospin RNA II kit (Macherey-Nagel) according to the manufacturer's protocol. Equal amount of RNA was used for reverse transcription (RevertAid H Minus First Strand cDNA Synthesis Kit, Thermo Scientific). qRT–PCR was performed with Light cycler 480 Real-Time PCR System (Roche) using Maxima SYBR Green qPCR Master Mix (2×) (Fermentas). At least three biological replicates were used for each genotype and at least three technical replicates were used for each biological replicate. Primer sequences are available upon request.

## Metabolite assays

Metabolite assays were performed from pre-wandering $3^{rd}$ instar larvae. All analyses were done at least in three biological replicates. Freshly hatched $1^{st}$ instar larvae were grown at controlled density (30/vial) on standard *Drosophila* medium. Samples were extracted from pre-wandering $3^{rd}$ instar larvae (90–96 h) and TAG assays were conducted using the coupled colorimetric assay (Triglyceride Reagent—Sigma; 82449, Free Glycerol Reagent—Sigma; F6428) and normalized to body weight as described previously (Tennessen *et al*, 2014). Hemolymph from third instar larvae was extracted as described previously (Zhang *et al*, 2011) and the circulating glucose was measured with the Glucose HK assay reagent (Sigma; GAHK-20) as described elsewhere (Tennessen *et al*, 2014).

## Fatty acid analysis by gas chromatography–mass spectrometry

Frozen third instar larvae were homogenized in 1% sulfuric acid in methanol and transmethylated by heating under nitrogen atmosphere (using hexane as cosolvent). The fatty acid methyl esters (FAME) formed were extracted with hexane in two steps. The dried and concentrated FAME were analyzed by a Shimadzu GC-2010 Plus gas chromatograph (Shimadzu, Kyoto, Japan) employing a ZBWAX capillary column (length 30 m, internal diameter 0.32 mm, film thickness 0.25 μm; Phenomenex, Torrance, CA, USA) and a flame ionization detector (FID). For the temperature-programmed runs, the sample solutions (2 μl) were injected in split mode, and helium was used as the carrier gas. The integrated and manually corrected (GCsolution software, Shimadzu) peak areas were converted to mol % by using the theoretical response factors for FID (Ackman, 1992) and calibrations with quantitative authentic standards (Supelco, Bellefonte, PA, USA). The FAME were identified by their mass spectra recorded by using a Schimadzu GCMS-QP2010 Ultra with mass selective detector (MSD). The results represent the FA composition of the total lipids.

## Lipid labeling and analysis of newly synthesized TAG by mass spectrometry

Freshly hatched L1 larvae (CG>Control and CG>ERK7) were grown at equal density (30/vial) on standard *Drosophila* medium. Early third instar larvae (72 h AED) were transferred to 10% yeast + 1% glucose or 10% yeast + 1% [$^{13}$C]glucose medium for 24 h. A total of 15 fat bodies per replicate were dissected, cellular lipids were extracted according to the Folch procedure (Folch *et al*, 1957) and

dissolved in chloroform/methanol 1:2. Immediately before mass spectrometry, 2% $NH_4OH$ was added along with an internal standard mixture. The samples were injected into a triple quadrupole mass spectrometer (Agilent 6490 Triple Quad LC/MS with iFunnel Technology; Agilent Technologies, Santa Clara, CA) at a flow rate of 10 μl/min, and spectra were recorded. TAGs were detected as $(M + NH_4)+$ ions (Duffin *et al*, 1991). Mass spectra were processed by MassHunter software (Agilent Technologies, Inc. California, USA) and the intensity proportions from unlabeled and mass-shifted [$^{13}$C]TAG ions were resolved by comparing the spectra of the samples with or without labeling. The total concentrations relating to the calculated intensity of the labeled (mmol [$^{13}$C] TAG produced in the pooled fat body sample in 24 h) and unlabeled TAG species were quantified against the internal standard TAG 54:3 (Sigma, T7140 Glyceryl trioleate). PEs were selectively detected using a head-group specific MS/MS scanning mode for neutral loss of 141 (NL141) and quantified against internal standards PE 28:2 and PE 40:2 (both synthesized in-house). To detect TAG accumulation after and before the 24-h [$^{13}$C]glucose feeding, the molar ratios of [$^{13}$C]TAG/PE and unlabeled TAG/PE ratios were calculated.

## Immunostainings and microscopy

Immunostainings were performed as described previously (Hasygar & Hietakangas, 2014); samples were mounted using Vectashield Mounting Medium with DAPI (Mediq) and imaged using a Leica TCS SP5 MP SMD FLIM and Leica SP8 Up-right microscopes. Fluorescence and compartment size (nuclear and nucleolar) were quantified by ImageJ software (NIH). Antibodies used were as follows: rabbit anti-PWP1 (Casper *et al*, 2011), mouse anti-fibrillarin (Abcam ab4566), anti-rabbit Alexa fluor 568 (Invitrogen), anti-rabbit Alexa fluor 633 (Invitrogen), and anti-mouse Alexa fluor 647 (Invitrogen).

## LipidTOX staining

Fat bodies from third instar larvae were fixed in 4% formaldehyde for 30 min, washed thrice with PBS. Fat bodies were stained with LipidTOX (H34477, Thermo Fisher) 1:400 in PBS, for 30 min at room temperature, washed thrice with PBS, and mounted using Vectashield Mounting Medium with DAPI (Mediq). Samples were imaged using a Leica TCS SP5 MP SMD FLIM and Leica SP8 Up-right microscopes and images were processed using ImageJ software (NIH).

## Image quantification

For lipid droplet analysis, the middle slices were selected from confocal stacks of fat bodies, and the outlines of the cells were manually defined. Droplets size and number per cell were identified by the Analyze Particles function in FIJI (Schindelin *et al*, 2012). Quantification of nuclear size was done by manually selecting outlines of individual cells at their maximum size in FIJI. For nucleolar size determination, same process was done with a selection of the maximum nucleolus size of a cell. Mean intensity data for region of interests were used to quantify subcellular localization changes of PWP1 detected by immunostaining. While the PWP1

signal from the whole nucleus was used to quantify the average immunofluorescence in the nucleus, immunofluorescence from five cytoplasmic regions within a cell was used to assess the cytoplasmic PWP1 signal.

### Cell culture and transfections

Culturing and transfections of *Drosophila* S2 cells were performed as described in (Liu *et al*, 2017).

### Western blotting

Phos-tag SDS–PAGE gels were made according to the manufacturer's protocol using 30 μM phos-tag (Wako Chemical). Samples were resolved on SDS–PAGE with or without Phos-tag and detected via Western blotting and bands were quantified using Image studio lite software (Li-COR).

### Pupal volume measurement

Pupal volume was quantified as previously described (Delanoue *et al*, 2010).

### Statistics

The statistical significances for the survival analysis experiments and pupariation rate experiments were analyzed by using the Log-rank test (JMP Pro 13 software). Statistical significances for rest of the experiments were determined by using unpaired Student's *t*-test. All quantitative data are presented as average ± standard deviation. No outliers were excluded.

## Data availability

The RNAseq data is available in GEO, with the accession number: GSE123901 (https://www.ncbi.nlm.nih.gov/geo/query/acc.cgi?acc= GSE123901).

**Expanded View** for this article is available online.

## Acknowledgements
We thank Catherine Rabouille, and Mark Van Doren for providing constructs and antibodies. Bohdana Rovenko and Richard Melvin are thanked for help with statistical analyses and other members of the Hietakangas laboratory, as well as Minna Poukkula, for feedback. We thank Juhana Juutila, Heini Lassila, Sanna Sihvo, DNA Sequencing and Genomics core facility and the Light Microscopy Unit of the Institute of Biotechnology for technical help and advice. *Drosophila* work was supported by Hi-Fly core facility. Funding was provided by the Academy of Finland (project grant 286767 & MetaStem Center of Excellence funding 312439 to VH), Sigrid Juselius Foundation (VH), Novo Nordisk Foundation (NNF16OC0021460, NNF18OC0034406, and NNF19OC0057478) to VH), The Diabetes Research Foundation (VH, KH), Integrative Life Science Doctoral Program (KH), Finnish Academy of Science and Letters (KH), Biomedicum Helsinki Foundation (KH), The Ella and Georg Ehrnrooth Foundation (KH), and Cancer Society of Finland (KH).

## Author contributions
Conceptualization: KH, VH; Formal analysis: KH, OD, JG, HR, KK; Funding acquisition: KH, VH; Investigation: KH, OD, YL, RH, HR; Methodology: RK; Project administration: VH; Visualization: KH, OD, JG, HR; Writing—original draft: KH, OD, HR, RK, VH; Writing—review and editing: KH, VH.

## Conflict of interest
The authors declare that they have no conflict of interest.

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
