## [Review Process File · EMBO Reports]

Coordinated control of adiposity and growth by anti-anabolic kinase ERK7

Kiran Hasygar, Onur Deniz, Ying Liu, Josef Gullmets, Riikka Hynynen, Hanna Ruhanen, Krista Kokki, Reijo Käkelä, and Ville Hietakangas

DOI: [10.15252/embr.201949602](https://doi.org/10.15252/embr.201949602)

Corresponding author(s): *Ville Hietakangas* (ville.hietakangas@helsinki.fi)

Review Timeline:

Submission Date:	5th Nov 19
Editorial Decision:	6th Dec 19
Revision Received:	30th Sep 20
Editorial Decision:	16th Nov 20
Revision Received:	18th Nov 20
Accepted:	27th Nov 20

Editor: Deniz Senyilmaz Tiebe

Transaction Report:

Dear Ville,

Thank you for the submission of your research manuscript to our journal, which was now seen by two referees, whose reports are copied below.

As you can see, the referees express interest in the proposed role of ERK7 in regulation lipid metabolism in flies. However, they also raise a number of concerns that need to be addressed to consider publication here. For instance, they request additional rescue and knock down experiments to increase conclusiveness of the findings. I find the reports informed and constructive, and believe that addressing the concerns raised will significantly strengthen the manuscript.

Given these constructive comments, we would like to invite you to revise your manuscript with the understanding that the referee concerns (as in their reports) must be fully addressed and their suggestions taken on board. Please address all referee concerns in a complete point-by-point response. Acceptance of the manuscript will depend on a positive outcome of a second round of review. It is EMBO reports policy to allow a single round of revision only and acceptance or rejection of the manuscript will therefore depend on the completeness of your responses included in the next, final version of the manuscript.

Supplementary/additional data: The Expanded View format, which will be displayed in the main HTML of the paper in a collapsible format, has replaced the Supplementary information. You can submit up to 5 images as Expanded View. Please follow the nomenclature Figure EV1, Figure EV2 etc. The figure legend for these should be included in the main manuscript document file in a section called Expanded View Figure Legends after the main Figure Legends section. Additional Supplementary material should be supplied as a single pdf labeled Appendix. The Appendix includes a table of content on the first page with page numbers, all figures and their legends. Please follow the nomenclature Appendix Figure Sx throughout the text and also label the figures according to this nomenclature. For more details please refer to our guide to authors.

2) individual production quality figure files as .eps, .tif, .jpg (one file per figure).

3) a .docx formatted letter INCLUDING the reviewers' reports and your detailed point-by-point responses to their comments. As part of the EMBO Press transparent editorial process, the point-by-point response is part of the Review Process File (RPF), which will be published alongside your paper. For more details on our Transparent Editorial Process, please visit our website:

<https://www.embopress.org/page/journal/14693178/authorguide#transparentprocess>

4) a complete author checklist, which you can download from our author guidelines (<http://embor.embopress.org/authorguide>). Please insert information in the checklist that is also reflected in the manuscript. The completed author checklist will also be part of the RPF.

5) Please note that all corresponding authors are required to supply an ORCID ID for their name upon submission of a revised manuscript (<https://orcid.org/>). Please find instructions on how to link your ORCID ID to your account in our manuscript tracking system in our Author guidelines (<http://embor.embopress.org/authorguide>).

6) We replaced Supplementary Information with Expanded View (EV) Figures and Tables that are collapsible/expandable online. A maximum of 5 EV Figures can be typeset. EV Figures should be cited as 'Figure EV1, Figure EV2' etc... in the text and their respective legends should be included in the main text after the legends of regular figures.

- For the figures that you do NOT wish to display as Expanded View figures, they should be bundled together with their legends in a single PDF file called *Appendix*, which should start with a short Table of Content. Appendix figures should be referred to in the main text as: "Appendix Figure S1, Appendix Figure S2" etc. See detailed instructions regarding expanded view here: <http://embor.embopress.org/authorguide#expandedview>.

7) We would also encourage you to include the source data for figure panels that show essential data.

Numerical data should be provided as individual .xls or .csv files (including a tab describing the data). For blots or microscopy, uncropped images should be submitted (using a zip archive if multiple images need to be supplied for one panel). Additional information on source data and instruction on how to label the files are available <http://embor.embopress.org/authorguide#sourcedata>.

8) Our journal encourages inclusion of *data citations in the reference list* to directly cite datasets that were re-used and obtained from public databases. Data citations in the article text are distinct from normal bibliographical citations and should directly link to the database records from which the data can be accessed. In the main text, data citations are formatted as follows: "Data ref: Smith et al, 2001" or "Data ref: NCBI Sequence Read Archive PRJNA342805, 2017". In the Reference list, data citations must be labeled with "[DATASET]". A data reference must provide the database name, accession number/identifiers and a resolvable link to the landing page from which the data can be accessed at the end of the reference. Further instructions are available at <http://embor.embopress.org/authorguide#datacitation>.

9) Regarding data quantification, please ensure to specify the name of the statistical test used to generate error bars and P values, the number (n) of independent experiments underlying each data point (not replicate measures of one sample), and the test used to calculate p-values in each figure legend. Discussion of statistical methodology can be reported in the materials and methods section, but figure legends should contain a basic description of n, P and the test applied. Please note that error bars and statistical comparisons may only be applied to data obtained from at least three independent biological replicates. Please also include scale bars in all microscopy images.

I look forward to seeing a revised version of your manuscript when it is ready. Please let me know if you have questions or comments regarding the revision.

Kind regards,

Deniz

Deniz Senyilmaz Tiebe, PhD
Editor
EMBO Reports

Referee #1:

The study by Hasygar and colleagues use a genetic approach to examine the role of the atypical MAP kinase ERK7 in *Drosophila* larval growth, developmental timing, and TAG accumulation. Using CRISPR/Cas9, they generated a loss of function mutation that induced elevated TAG levels. Moreover, fat body specific RNAi induced a similar phenotype. In contrast, ERK7 overexpression induced a lean phenotype. The authors then observe that ERK7 mutants display changes in PWP1 localization/phosphorylation and demonstrate that PWP1 also regulates TAG accumulation. Finally, they link ERK7 activity with expression of the transcription factor sugarbabe, which promotes lipid storage.

Overall, I find the study potentially interesting because it describes a new mechanism regulating lipid metabolism in the fly. My only request is that the authors conduct the a few rescue and epistasis experiments necessary to fully support their conclusions:

(1) Attempt to rescue the ERK7 mutant phenotype using the UAS-ERK7 transgenes.

(2) In lines 158-164, PWP1 RNAi induces a reduction in TAG levels, hinting at the possibility that PWP1 acts downstream of ERK7 to control fat accumulation. To further prove this hypothesis, the authors should try to rescue the ERK7 mutant TAG phenotype overexpressing PWP1 in the fat body.

(3) Similar to point 2, does overexpression of sugarbabe in the fat body of ERK7 mutants also rescue the TAG phenotype?

I also have concerns with the conclusion that ERK7 is directly regulating PWP1 phosphorylation.

This is suggested in both the results (line 153) and discussion. Since there is no data in the manuscript demonstrating that ERK7 phosphorylates PWP1, the authors should make more nuanced statements.

Additional Concerns:

Line 90: The authors suggest that overexpression of ERK7 in fat body clones results in both smaller cell size and decreased lipid droplet size. While the decrease in cell size is apparent, I see no decrease in lipid droplet size within this figure panel. Either the authors should provide quantitative data that demonstrates a decrease in lipid droplet size or remove this claim from the paper, as I don't think it's important for their argument.

Line 124 and 125: This starvation experiment indicates that ERK7 is required to arrest growth upon nutrient deprivation, but key details of the experiment are missing. What stage are the animals? Are they pre- or post-critical weight?

Line 124 and 125: What is the nutrient deprivation media? Is it the same as the fasting media described in the methods?

Referee #2:

In this MS, Hasygar and cols study the role of ERK7 in *Drosophila*. The authors generate an ERK7 null mutant strain. These mutants show increased TAG levels and faster growth when raised in rich food, and are sensitive to nutrient deprivation. ERK7 manipulation affects the localization of PWP1, which in turn represses sugarbabe and therefore controls the expression of biosynthetic elements and ribosomal RNA, limiting TAG storage.

The results presented in this MS are potentially interesting but the authors need to address some issues before it is ready for publication. In general, the authors base most of their conclusions in ERK7 gain of function experiments. Showing the result of ERK7 loss of function will be required to prove that the interactions identified in this MS are physiologically relevant and the phenotypic outcomes are not due to non-specific defects consequence of gene hyperactivation.

Specific concerns below:

To rule out potential non-specific effects of the UAS-ERK7-RNAi line used in this study, the authors should reproduce the results shown here with independent UAS-RNAi lines.

Fig 1E shows a GFP clone overexpressing ERK7. The authors claim that this cells shows a reduction in the number of lipid droplets. Even though that's true, it is also true that the cell size is dramatically reduced. The authors should show this observation in a quantified manner. They should generate multiple clones and normalize the number of droplets to the cell size. They should also show the number of droplets in ERK7 mutants or UAS-RNAi expressing clones. Otherwise, the effect in the over-expression experiment might not have any physiological relevance and could be due an unspecific effect in due to gene over-expression. For the same reason, the authors should show the expression of genes presented in Fig 1 in ERK7 mutants.

Showing the graphs in Fig 1A-C, and F1, G, and 5C, E, using the same scale would allow for better comparison between the different results obtained in the different experiments.

The authors should analyze fatty acids composition of total lipids (Fig 2) in the ERK7 mutant background. The same applies to the results shown in Fig 4, Fig 5A-C.

Line 139, "The nuclei of the ERK7-expressing cells were significantly smaller in size, which suggests impaired growth through endoreplication (Figure 4A, B)." Why? Impaired growth might be due to other processes. Endoreplication should instead lead to an increase in DNA content and therefore bigger nuclear size, right? The authors should explain and clarify this issue.

Does Sugarbabe expression rescue fat body cell size and lipid droplet number in ERK7-expressing clones in the fat body?

Point-by-point response, Hasygar et al.

We thank the reviewers for the constructive feedback that has helped us to improve the manuscript. The main points of critique were on insufficient genetic epistasis data (reviewer #1) and basing the conclusion too much on gain-of-function data (reviewer #2). Taking both viewpoints into account (and the limitations posed by the Covid-19 lab restrictions), we have now conducted a series of experiments to further test our conclusions. As detailed below, loss-of-function experiments now provide additional evidence for 1) the role of ERK7 as a negative regulator of lipid storage, 2) the role of ERK7 in regulating lipid metabolism gene expression, 3) the regulation of PWP1 localization by ERK7, and 4) the regulation of nuclear and nucleolar size by ERK7. Moreover, genetic epistasis experiments (performed by loss-of-function) provide further support for our model that PWP1 and Sugarbabe act downstream of ERK7 to coordinate lipid storage and growth.

Following these revisions (as well as others) we now conclude that the data presented in the revised manuscript should sufficiently support the conclusion that ERK7 negatively regulates growth and lipid storage in a coordinated manner in the *Drosophila* fat body and mediates these functions (at least in part) through the regulation of PWP1 and Sugarbabe.

Referee #1:

The study by Hasygar and colleagues use a genetic approach to examine the role of the atypical MAP kinase ERK7 in *Drosophila* larval growth, developmental timing, and TAG accumulation. Using CRISPR/Cas9, they generated a loss of function mutation that induced elevated TAG levels. Moreover, fat body specific RNAi induced a similar phenotype. In contrast, ERK7 overexpression induced a lean phenotype. The authors then observe that ERK7 mutants display changes in PWP1 localization/phosphorylation and demonstrate that PWP1 also regulates TAG accumulation. Finally, they link ERK7 activity with expression of the transcription factor sugarbabe, which promotes lipid storage.

Overall, I find the study potentially interesting because it describes a new mechanism regulating lipid metabolism in the fly. My only request is that the authors conduct the a few rescue and epistasis experiments necessary to fully support their conclusions:

(1) Attempt to rescue the ERK7 mutant phenotype using the UAS-ERK7 transgenes.

*We attempted to perform the rescue experiment as suggested. Unfortunately, the double GFP-balanced stocks with the mutant and transgene, that are needed for this experiment, were not viable. To further confirm the ERK7 loss-of-function phenotypes, we have now conducted experiments with fat body-specific ERK7 knockdown using a second independent RNAi line, which reproduced our original findings on TAG storage and growth. The new data is shown in **Figures EV2A and EV3B** of the revised manuscript.*

In conclusion, we now have demonstrated that ERK7 inhibits TAG accumulation and growth in vivo by using four independent genetic approaches: mutant, 2 non-overlapping RNAi lines, and overexpression (specificity of which was verified by kinase-dead point mutants).

(2) In lines 158-164, PWP1 RNAi induces a reduction in TAG levels, hinting at the possibility that PWP1 acts downstream of ERK7 to control fat accumulation. To further prove this hypothesis, the authors should try to rescue the ERK7 mutant TAG phenotype overexpressing PWP1 in the fat body.

Overexpression of PWP1 is not expected to rescue ERK7 mutant TAG phenotype, as ERK7 mutant has elevated TAG levels and PWP1 is necessary for maintaining TAG levels (PWP1 loss-of-function downregulates TAG).

*However, to test genetic epistasis between ERK7 and PWP1 we have used simultaneous knockdown of ERK7 and PWP1 in the fat body. Consistent with our earlier data, ERK7 RNAi in the fat body increased the number of lipid droplets per cell. Interestingly, PWP1 loss-of-function suppressed the increase in lipid droplets. Similar suppression was observed with the nuclear size phenotype. Our new data (**Figures 6G, 6H and 6I**) is consistent with the model that PWP1 acts downstream of ERK7 to control lipid storage and growth.*

(3) Similar to point 2, does overexpression of sugarbabe in the fat body of ERK7 mutants also rescue the TAG phenotype?

Overexpression of Sugarbabe is not expected to rescue ERK7 mutant TAG phenotype, as ERK7 mutant has elevated TAG levels and Sugarbabe overexpression increases TAG levels. We have earlier shown that Sugarbabe overexpression partially rescues TAG levels upon ERK7 overexpression.

*To further test genetic epistasis between ERK7 and Sugarbabe, we have analyzed lipid droplets in ERK7/sugarbabe double mutants. Our new data shows that hypomorphic mutant of sugarbabe suppresses the increase in the number of lipid droplets observed in ERK7 mutants. This further strengthens our conclusion that sugarbabe acts downstream of ERK7 to control lipid storage. Moreover, our data shows that loss of sugarbabe also suppresses the increase in nuclear size in ERK7 mutant. The new data is presented as **Figures 7D, 7E, and 7G** of the revised manuscript.*

I also have concerns with the conclusion that ERK7 is directly regulating PWP1 phosphorylation. This is suggested in both the results (line 153) and discussion. Since there is no data in the manuscript demonstrating that ERK7 phosphorylates PWP1, the authors should make more nuanced statements.

As suggested, we have modified our conclusion about PWP1 phosphorylation, stating that the effect can be either direct or indirect (lines 190-191).

Additional Concerns:

Line 90: The authors suggest that overexpression of ERK7 in fat body clones results in both smaller cell size and decreased lipid droplet size. While the decrease in cell size is apparent, I see no decrease in lipid droplet size within this figure panel. Either the authors should provide quantitative data that demonstrates a decrease in lipid droplet size or remove this claim from the paper, as I don't think it's important for their argument.

*As suggested by the Reviewer, we have now quantified the lipid droplet data for ERK7 overexpressing clones. Our results show both significantly reduced lipid droplet numbers and reduced lipid droplet average size in ERK7 overexpressing clones. The new data is shown in **Figures 1I and 1J** of the revised manuscript.*

Line 124 and 125: This starvation experiment indicates that ERK7 is required to arrest growth upon nutrient deprivation, but key details of the experiment are missing. What stage are the animals? Are they pre- or post-critical weight?

The larvae were 72 h old (AED), i.e. pre-critical weight. We have now modified the Materials & Methods accordingly (line 351).

Line 124 and 125: What is the nutrient deprivation media? Is it the same as the fasting media described in the methods?

Yes. The experimental conditions are described in the Materials & Methods under “Fasting conditions”. We have changed the wording in Results (line 162) to avoid confusion.

Referee #2:

In this MS, Hasygar and cols study the role of ERK7 in Drosophila. The authors generate an ERK7 null mutant strain. These mutants show increased TAG levels and faster growth when raised in rich food, and are sensitive to nutrient deprivation. ERK7 manipulation affects the localization of PWP1, which in turn represses sugarbabe and therefore controls the expression of biosynthetic elements and ribosomal RNA, limiting TAG storage.

The results presented in this MS are potentially interesting but the authors need to address some issues before it is ready for publication. In general, the authors base most of their conclusions in ERK7 gain of function experiments. Showing the result of ERK7 loss of function will be required to prove that the interactions identified in this MS are physiologically relevant and the phenotypic outcomes are not due to non-specific defects consequence of gene hyperactivation.

The Reviewer raises the concern about non-specific defects caused by gene hyperactivation leading to physiologically irrelevant findings. While such a risk certainly exists, non-specific phenotypes of proteins are most commonly dominant negative effects (PMID: 23583758), which were earlier ruled out by using kinase-dead mutants as control. Nevertheless, we have now conducted several new experiments by using various loss-of-function strategies. The main conclusions of our study are now supported by several lines of loss- and gain-of-function data.

Specific concerns below:

To rule out potential non-specific effects of the UAS-ERK7-RNAi line used in this study, the authors should reproduce the results shown here with independent UAS-RNAi lines.

*To confirm the loss-of-function data further, we have now reproduced the larval growth and TAG phenotypes using a second independent RNAi line. The new data is shown in **Figures EV2A and EV3B** of the revised manuscript.*

Fig 1E shows a GFP clone overexpressing ERK7. The authors claim that this cells shows a reduction in the number of lipid droplets. Even though that's true, it is also true that the cell size is dramatically reduced. The authors should show this observation in a quantified manner. They should generate multiple clones and normalize the number of droplets to the cell size.

*We have now quantified the lipid droplets as requested by the Reviewer. Our data shows that both lipid droplet number and lipid droplet mean size are significantly reduced by ERK7 overexpression (**Figures 1I and 1J**). We have also normalized the lipid droplet total volume to the cell size, as requested. ERK7 overexpression causes significant reduction in lipid droplet total volume, even when normalized in such a way (**Figure EV2D**). This implies that there is a growth independent contribution of ERK7 on lipid storage. This conclusion is further supported by the new data showing increase in TAG/PE ratio in ERK7 mutants (**Figure 1F**).*

Despite these findings, however, it is not our intention to conclude that the regulation of fat body cell size and lipid storage by ERK7 are primarily parallel functions, but rather they seem to be

regulated in a highly coordinated manner. As a net consequence of this coordinated regulation, the TAG storage of the animal is affected (as shown by the biochemical measurements of TAG levels). The conclusion on coordinated regulation of lipid storage and growth is further supported by the new data showing that the loss of PWP1 and Sugarbabe suppress both phenotypes in ERK7 loss-of-function animals. These findings are in line with data of other known regulators of growth and metabolism, such as mTOR, which is known to coordinate growth and lipid synthesis (e.g. see PMID: 18762023).

They should also show the number of droplets in ERK7 mutants or UAS-RNAi expressing clones. Otherwise, the effect in the over-expression experiment might not have any physiological relevance and could be due an unspecific effect in due to gene over-expression.

*As requested, we have now quantified lipid droplet numbers in ERK7 mutants and in ERK7 RNAi expressing fat bodies. In line with the previous ERK7 mutant and RNAi TAG data, ERK7 loss-of-function fat bodies show elevated number of lipid droplets per cell. The new data is shown in **Figures 1D, 1E, 6G and 6H** of the revised manuscript.*

For the same reason, the authors should show the expression of genes presented in Fig 1 in ERK7 mutants.

*Out of the three genes shown in original Figure 1 only Lipase 4 was significantly deregulated in ERK7 mutants. Therefore, we have chosen to broaden the gene expression analysis. We are now displaying new data of an RNA sequencing experiment in the ERK7 mutants. Unbiased gene set enrichment analysis shows significant upregulation of GO terms 'lipid particle' and 'regulation of lipid storage' in ERK7 mutants. Individual genes involved in lipid biosynthesis, such as Seipin, Agpat3, ATPCL as well as lipid chaperone Fabp show upregulation in ERK7 mutants and a corresponding downregulation upon ERK7 overexpression. Conversely, expression of Lipase 4 and Triglyceride lipase CG34448 show downregulation in ERK7 mutants and corresponding upregulation upon ERK7 overexpression. The new gene expression data is shown as **Figure 3** of the revised manuscript. Collectively, the gene expression data is consistent with the conclusion of ERK7 being a negative regulator of TAG storage.*

Showing the graphs in Fig 1A-C, and F1, G, and 5C, E, using the same scale would allow for better comparison between the different results obtained in the different experiments.

Has been modified, as suggested, whenever feasible.

The authors should analyze fatty acids composition of total lipids (Fig 2) in the ERK7 mutant background.

Our ability to conduct lipidomics experiments was severely compromised by the Covid-19-related lab restrictions. We were only able to conduct one simple lipidomics experiment to show specific increase in TAG/PE ratio (see the point above). Therefore, we have now modified the Results section regarding the fatty acid composition experiment to emphasize it was performed by overexpression (line 118). We think that the fatty acid analysis is not essential for our main conclusion (ERK7 is a negative regulator of lipid storage). Even though not verified by loss-of-function data, we would prefer including the fatty acid composition data into the manuscript, as it might be valuable for readers with specific interest in lipid metabolism. However, we are prepared to omit the data, if the reviewers and editor consider it necessary.

The same applies to the results shown in Fig 4, Fig 5A-C.

As requested, we have now analyzed the nuclear size and nucleolar area/nuclear area ratio in ERK7 mutants. Consistent with our model, ERK7 mutant fat bodies show increased nuclear size

and increased nucleolar/nuclear area ratio. The new data is now shown in **Figures 5A, 5B, and 5C** of the amended manuscript. This data further supports the conclusion that ERK7 is a negative regulator of growth in the fat body.

As also requested, we have analyzed the subcellular localization of PWP1 in ERK7 mutants. Interestingly, our data shows increased nucleolar localization of PWP1 in ERK7 mutants. This is in line with the increased growth observed in ERK7 mutants. The new data is shown as **Figures 6B and 6C** of the amended manuscript. This data further supports the conclusion that ERK7 regulates PWP1 localization.

Line 139, "The nuclei of the ERK7-expressing cells were significantly smaller in size, which suggests impaired growth through endoreplication (Figure 4A, B)." Why? Impaired growth might be due to other processes. Endoreplication should instead lead to an increase in DNA content and therefore bigger nuclear size, right? The authors should explain and clarify this issue.

This was indeed written in a confusing manner. We have now modified the text (line 177).

Does Sugarbabe expression rescue fat body cell size and lipid droplet number in ERK7-expressing clones in the fat body?

*As the main critique was on relying too strongly on ERK7 overexpression, we have chosen to perform the suggested epistasis experiment by using a loss-of-function strategy, with ERK7 and Sug mutants. This experiment shows that loss of sugarbabe suppresses the increased lipid droplet number in ERK7 mutants (as well as the increase in nuclear size). The new data is shown as **Figures 7D, 7E, and 7G** of the revised manuscript. The data is consistent with the conclusion that Sugarbabe controls lipid storage and growth downstream of ERK7.*

Dear Ville,

Thank you for submitting your revised manuscript. It has now been seen by both of the original referees.

As you can see, the referees find that the study is significantly improved during revision and recommend publication. Before I can accept the manuscript, I need you to address some minor points below:

- Please provide 3-5 keywords for your study. These will be visible in the html version of the paper and on PubMed and will help increase the discoverability of your work.
- Please add 'Conflict of Interests' and 'Author Contributions' sections.
- We notice the presence of 'data not shown' phrase on page 11, which is not allowed to be used as per journal policy. Please either show the data, or remove the statement.
- Please make sure that funding information is complete both in the manuscript and the manuscript submission system - we note that currently they don't match.
- Papers published in EMBO Reports include a 'synopsis' and 'bullet points' to further enhance discoverability. Both are displayed on the html version of the paper and are freely accessible to all readers. The synopsis includes a short standfirst summarizing the study in 1 or 2 sentences that summarize the paper and are provided by the authors and streamlined by the handling editor. I would therefore ask you to include your synopsis blurb and 3-5 bullet points listing the key experimental findings.
- In addition, please provide an image for the synopsis. This image should provide a rapid overview of the question addressed in the study but still needs to be kept fairly modest since the image size cannot exceed 550x400 pixels.
- Our production/data editors have asked you to clarify several points in the figure legends (see attached document). Please incorporate these changes in the attached word document and return it with track changes activated. I am aware that the comments were made on an earlier version of the manuscript, please use the attached document as a reference and perform the changes on the latest version of the text.

Thank you again for giving us to consider your manuscript for EMBO Reports, I look forward to your minor revision.

Kind regards,

Deniz

--

Deniz Senyilmaz Tiebe, PhD
Editor
EMBO Reports

Referee #1:

The authors have addressed my concerns

Referee #2:

The authors have addressed all my concerns and the paper is ready for publication in its current version.

The authors have addressed all minor editorial requests.

Dear Ville,

Thank you for submitting your revised manuscript. I have now looked at everything and all is fine. Therefore, I am very very pleased to accept your manuscript for publication in EMBO Reports.

Congratulations on a nice work!

Kind regards,

Deniz

--

Deniz Senyilmaz Tiebe, PhD

Editor

EMBO Reports

--

At the end of this email I include important information about how to proceed. Please ensure that you take the time to read the information and complete and return the necessary forms to allow us to publish your manuscript as quickly as possible.

As part of the EMBO publication's Transparent Editorial Process, EMBO reports publishes online a Review Process File to accompany accepted manuscripts. As you are aware, this File will be published in conjunction with your paper and will include the referee reports, your point-by-point response and all pertinent correspondence relating to the manuscript.

If you do NOT want this File to be published, please inform the editorial office within 2 days, if you have not done so already, otherwise the File will be published by default [contact: emboreports@embo.org]. If you do opt out, the Review Process File link will point to the following statement: "No Review Process File is available with this article, as the authors have chosen not to make the review process public in this case."

Should you be planning a Press Release on your article, please get in contact with emboreports@wiley.com as early as possible, in order to coordinate publication and release dates.

Thank you again for your contribution to EMBO reports and congratulations on a successful publication. Please consider us again in the future for your most exciting work.

THINGS TO DO NOW:

You will receive proofs by e-mail approximately 2-3 weeks after all relevant files have been sent to our Production Office; you should return your corrections within 2 days of receiving the proofs.

Please inform us if there is likely to be any difficulty in reaching you at the above address at that time. Failure to meet our deadlines may result in a delay of publication, or publication without your corrections.

All further communications concerning your paper should quote reference number EMBOR-2019-49602V3 and be addressed to emboreports@wiley.com.

Should you be planning a Press Release on your article, please get in contact with emboreports@wiley.com as early as possible, in order to coordinate publication and release dates.

Corresponding Author Name: Ville Hietakangas

Manuscript Number: EMBOR-2019-49602V1